# Active Flooding Mitigation for Stability Enhancement in a Damaged RoPax Ship

**Petri Valanto**

Hamburg Ship Model Basin HSVA, Bramfelder Strasse 164, D-22305 Hamburg, Germany; valanto@hsva.de

**Abstract:** In the framework of the EU project Flooding Accident Response (FLARE), flooding mitigation on a RoPax ship was studied using different active methods to improve ship safety in damage cases leading to a hull breach and flooding. Adding active flooding control systems to ship designs and ships in service, which would mitigate the effects of flooding in a damage case, could be an attractive way to improve ship safety. In order to promote this idea, the effects of such active measures on the ship safety were studied: the choice of them, the required application speed of them, their functionality in waves, the numerical modeling of them, and finally testing them with model scale tests. The following flooding mitigation methods were studied: (1) counter flooding, (2) the recovery of lost buoyancy in a damaged compartment, and (3) deploying a watertight barrier on the trailer deck. This study consists of the numerical simulations carried out with the program HSVA Rolls in chosen damage cases on a current RoPax design, with and without active flooding mitigation measures, and of the following ship model tests based on the a priori computations.

**Keywords:** ship damage stability; active flooding mitigation; numerical simulation; model tests

## 1. Introduction

In the framework of the EU project Flooding Accident Response (FLARE), flooding mitigation on a RoPax ship was studied as an active way to improve ship safety in damage cases leading to a hull breach and flooding. According to the present grandfather policy in IMO regulations, new rules and requirements apply in general only to new buildings, not least as such new requirements in damage stability tend to have effects on the ship subdivision. Changes in the subdivision of existing ships would be prohibitively costly. Another, easier way to elevate the safety level of both new buildings and existing ships would be to add active flooding control systems to the ship designs and ships, which would mitigate the effects of flooding in a damage case, see, e.g., reference [1] for an introduction. In order to promote this idea, the effects of such active measures on the ship safety were studied: the choice of them, the required application speed of them, their functionality in waves, the numerical modeling of them, and finally, testing them with model scale tests.

This study describes the numerical simulations carried out by the HSVA in chosen damage cases on a current RoPax design, with and without active flooding mitigation measures applied. The subsequent experimental verification of these mitigation methods and related computations are also described.

In order to study the mitigation methods, the behavior of the damaged RoPax vessel in calm water and in waves was simulated with the program HSVA Rolls [2–5] for both the passive cases and those with active flooding control. That is, each case was studied with and without flooding mitigation, allowing a good comparison. These numerical flooding investigations served also as an important preparation for the following model test campaign on mitigation. On few selected damage cases, model tests were carried out to verify the calculations and to demonstrate the benefits of active flooding control.

The simulations and tests in calm water provide information on the ability of the mitigation system to prevent a rapid capsize in a transient flooding and heeling phase during and just after the damage opening. For this, both the damage opening duration in time and the reaction of the mitigation system were modeled first in the numerical simulations and later in the HSVA model tests. The simulations and tests in irregular seas provide information on the ability of the ship to survive gradual flooding in beam seas in damaged conditions mostly after the flooding mitigation has taken place. All sea states in this investigation were modeled with JONSWAP-Spectrum, with the peak enhancement factor $\gamma$ of 3.3 and peak wave period $T_P$ of 10.0 s.

The RoPax ship under this study is a modern northern RoPax design made for research purposes only. The safety of this design was studied by various partners in the framework of FLARE at a few levels of sophistication in several damage cases. These were first screened with simpler methods and consequently six critical cases were further studied with forensic analysis using the HSVA Rolls as a simulation program. Two of these six damage cases (No. 2 and 4) were selected to be used in investigating the effectiveness of flooding mitigation with numerical simulations and model tests.

## 2. General Description of the RoPax Vessel

The ship used in the numerical simulations and model tests is a 162 m-long RoPax vessel design to the SOLAS 2020 standard by the Meyer Turku (MT) shipyard. The ship is designed as a day ferry, hosting up to 1900 passengers and a crew of 91. It has 800 m trailer lanes on the main trailer deck at 9.2 m above the baseline and 1050 m car lanes in the garage deck [6]. The main particulars of the vessel at the test draught are given in Table 1, and views of the ship and ship lines are given in Figures 1–3.

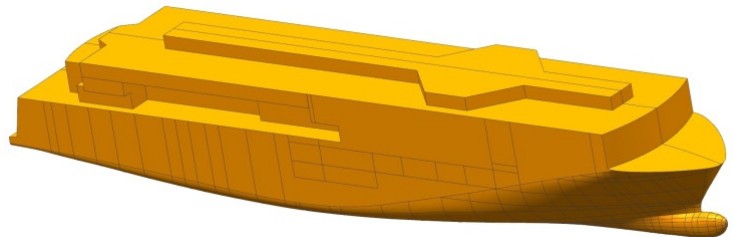

**Figure 1.** CAD model of the MT RoPax used in the numerical simulations of flooding mitigation.

**Table 1.** Main data of the vessel in intact (test) condition.

| MT RoPax—HSVA Model No: 5539 | Symbol | Unit | Ship |
|---|---|---|---|
| Length overall | $L_{OA}$ | m | 162.00 |
| Length between perpendiculars | $L_{PP}$ | m | 146.72 |
| Breadth at waterline | $B_{WL}$ | m | 28.00 |
| Draught at aft perpendicular | $T_A$ | m | 6.30 |
| Draught at forward perpendicular | $T_F$ | m | 6.30 |
| Depth to trailer deck | D | m | 9.20 |
| Displaced volume (bare hull) | $V_{BH}$ | m³ | 16799.4 |
| Block coefficient | $C_B$ | - | 0.6522 |
| Intact transverse GM | GM | m | 2.50/3.40 |

A version of the MT RoPax subdivision modified by the Maritime Safety Research Centre of the University of Strathclyde (MSRC) for the purposes of the initial screening of damage cases in FLARE and for further forensic analysis was used in the present investigation of the flooding mitigation efforts.

The ship model was built of reinforced flax fiber composite material in a scale of 1:28 according to the hull lines provided by the Meyer Turku Shipyard, but the bow and stern thruster tunnels were not modeled. For the model tests, the following appendages were attached to the model: rudders, propellers and shafts, struts, and bilge keels.

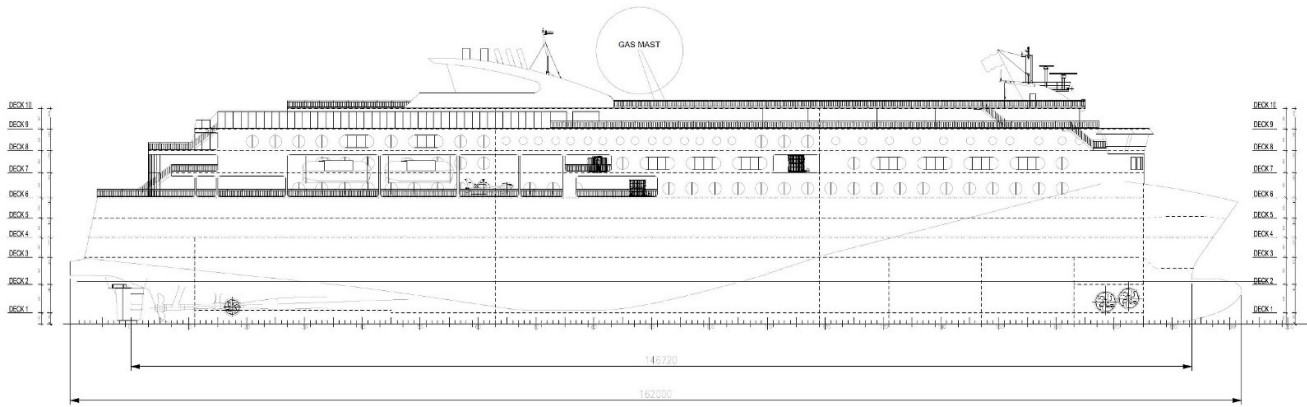

**Figure 2.** Side view of the MT RoPax design.

All ship and model particulars, as well as results, are presented in full scale (f.sc.). At the studied draft of 6.3 m, the minimum metacentric height (GM) value according to the current SOLAS Ch. II-1 requirements is 3.4 m. This ensures a good survivability level of the intact vessel. Therefore, also smaller GM values were used in the simulations and model tests to achieve capsize cases for testing the mitigation methods on the damaged vessel.

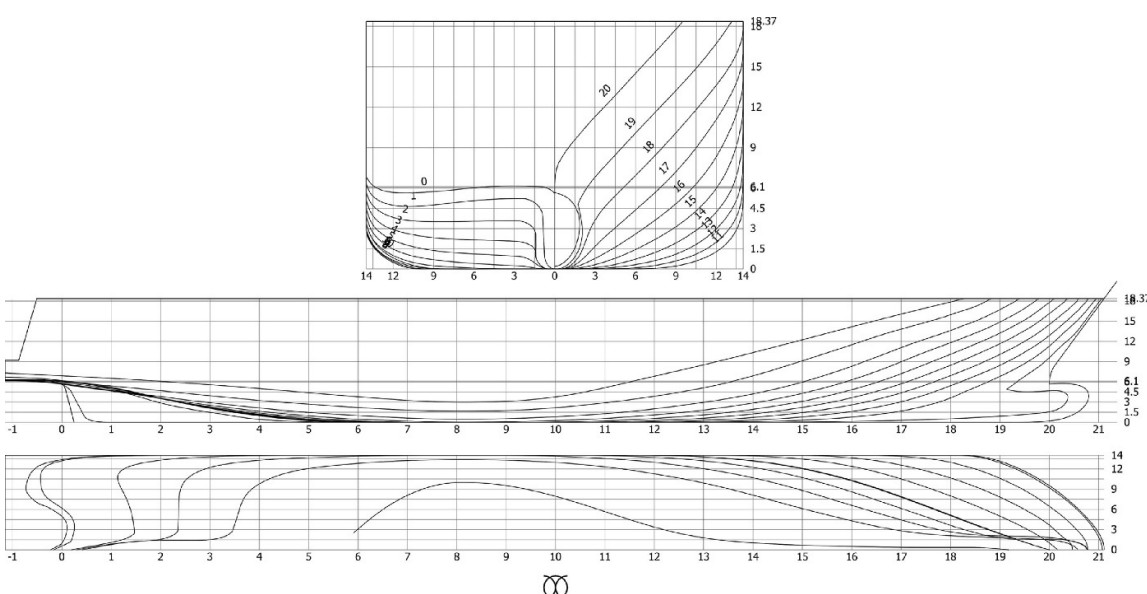

**Figure 3.** Bare hull lines drawing of the studied RoPax ship [7].

## 3. Collision Damage Opening Times

The damage cases under investigation comprise as loss mechanism both transient capsize and progressive flooding. In many of these cases, the damage opening time (DOT) can be relevant. For this reason, the damage cases were not studied with an instant damage opening (0 s), which in case of a collision damage is not realistic, but which as an approximation is sometimes applied to study transient damage cases. Instead, all cases were investigated using the damage opening time of 15 s, in addition to few cases with the longer

opening time of 30 s. The value 15 s is approximately the shortest realistic collision damage opening time for the MT RoPax. It is based on the assumption of the shortest time, at which the striking ship can pull itself back, with its damaged bow withdrawing out of the collision damage penetration on the struck ship, using its available propulsion power full astern (see Figure 4). The striking ship and the stuck ship are assumed identical. Such damage opening times were investigated by HSVA in FLARE and are given for several vessel types in reference [8]. The value 15 s based on the analysis represents a time in which ca. 90% of the damage opens. The damage opening rate is highly non-linear. Therefore, this representative effective value was chosen for the numerical simulations and model tests.

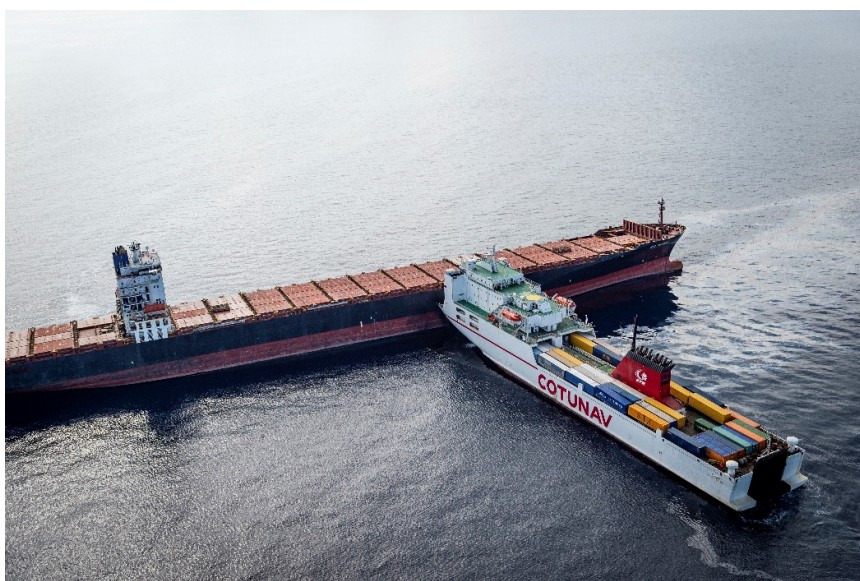

**Figure 4.** An example of a ship-to-ship collision in the Mediterranean Sea on 8 October 2018. The instant opening of the related collision damage is more a myth than a sound physical assumption. Photo: © picture-alliance/AP Photo/Marine Nationale/Benoit Emile.

## 4. Numerical Simulations and Model Tests on Flooding Mitigation

### 4.1. The Code HSVA Rolls

The survivability of damaged RoPax ships in case of a flooding accident is critical, as these ships have a tendency for a rapid capsize, often not allowing for an orderly evacuation. Various time-domain flooding simulation tools have been developed to study their behavior.

The in-house version of the commonly used German code Rolls [2–4], namely the HSVA Rolls, was used in this study [5]: Floodwater in internal compartments and on decks can be modelled either with shallow-water-equations (SWE) or with a pendulum model. For practically all cases in this study, SWEs were used to model the flow on the trailer deck, and the pendulum model was used for the more deeply flooded compartment spaces below. Flow rates through the breaches are based on Bernoulli's equation. For the ship heave, pitch, sway, and yaw motions the method uses response amplitude operators (RAO) determined in the frequency domain with a linear strip method. The roll and surge motions are determined with the time integration of the non-linear equations of motion coupled with the other four degrees of freedom. The hydrodynamic contributions are based on linear strip theory and of those based on the water motions in internal compartments. The hydrostatic contributions in calm water and waves are non-linear and are based on calculations with NAPA software. The trailer deck was discretized with a 160 × 30 SWE grid, resulting in altogether 3650 elements.

Such codes are used in the design of new, safer ships and more widely in research projects. Consequently, the validation and benchmarking of these tools are essential. The latest very recent benchmarking of the HSVA Rolls among other codes can be found in reference [7].

*4.2. Introduction to Numerical Simulations and Model Tests on Flooding Mitigation*

The damage cases on the RoPax under investigation consist of a breach to the trailer deck and to several damaged compartments below. When the damage opens, the latter will be flooded quite rapidly, and the trailer deck with small delay when the ship has heeled sufficiently.

There is a center casing on the port side adjacent to the centerline on the trailer deck, which has an impact on the accumulation of water on the deck in waves. Further, the flooding of the trailer deck is limited by additional transverse bulkheads present in the MSRC version of the ship subdivision used.

The arrangements of the floodable compartments for the damage cases studied are illustrated below in the following chapters. There are no internal connections between the compartments. In the physical ship model, all damaged compartments were ventilated through ventilation pipes in the compartment corners. Consequently, full ventilation was applied in the simulations. Due to the large scale (1:28) of the model, the openings are quite large, and therefore, the industry standard discharge coefficient 0.6 was used for all openings. The capsize criterion in this study is the heeling angle of 60°. The time to capsize (TTC) is based on this value.

The opening of the damage on the ship hull was modeled using a time-dependent discharge coefficient for the flow into the damaged compartments, with its value typically changing from 0 to 0.6 in the pre-set damage opening time.

The counter flooding (CF) was modeled by simply opening two counter flooding compartments to the sea through a change of the discharge coefficient value from 0 to 0.6 once a pre-set ship heeling angle was exceeded in the simulation. The recovery of lost buoyancy (RLB) by displacing floodwater in a damaged compartment was modeled with reducing the permeability of the compartment in a pre-set duration of time.

The deployment of the watertight barrier (WTB) on the trailer deck was modeled as follows: The floodwater motion was described with shallow-water-equations on a numerical grid spanning over the whole deck. Setting new boundary conditions during the simulation would be difficult. Instead, in four cell rows at the position of the deployable barrier, the horizontal fluid acceleration was artificially increased to keep water out of these cells (see Figure 18). This effectively keeps the barrier watertight after being lowered.

As the corresponding model testing techniques used were all new, a short description of these is given in Appendix A.

*4.3. Numerical Simulation of Damage Case 2 with and without Counter Flooding*

In Damage Case 2 (MSRC DMC0569; DC2), the compartments T100 (max. volume 357 m³) and T141 (777 m³) on the starboard side of the MT RoPax get damaged and are flooded, as illustrated in Figure 5. The damage opening extends also to the trailer deck, the floodable area of which is limited by two transverse bulkheads. The damage opening size is about 18.3 m in length and extends 1 m below the still water level at the ship draught of 6.3 m. Without mitigation, the ship capsizes with GM 2.5 m in about 70 s when the damage opens in 15 s. With GM 2.95 m, the ship survives in calm water, but in beam seas with significant wave height $H_s$ 2.0 m, it capsizes in ca. 75–180 s when damage is opened in 15 s. With the original design GM value of 3.40 m, the ship does not capsize in calm water or in beam seas until the significant wave height reaches 2.5 m. Thus, the mitigation efforts were studied using the lowest GM value of 2.5 m in calm water, and with damage opening times of 15 s and 30 s. In both conditions, the ship capsizes in calm water without mitigation. The damaged ship was further studied in beam seas with the GM value 3.4 m.

The mitigation effort consisted of flooding the compartments T042 (586 m³) and T099 (357 m³) on the undamaged port side of the ship. In the first test case, the damage opening time 15 s was used. The counter flooding in the compartments T042 and T099 was as-

sumed to start when the ship heeling angle to the damaged side exceeds 5°. In the numerical simulation the relatively small openings to counter flooding compartments were opened instantly in zero-time duration.

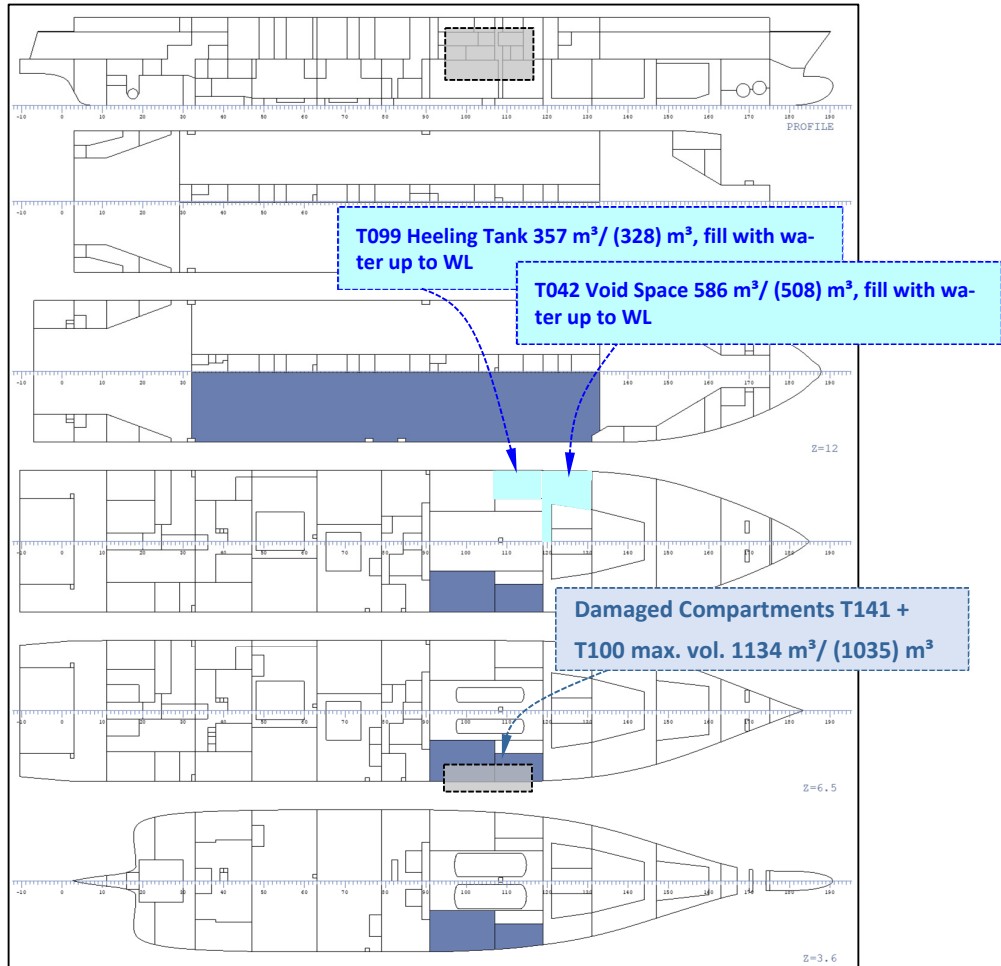

**Figure 5.** Damage Case 2 with counter flooding: The grey rectangle shows the damage penetration. The damaged compartments are shown with dark blue-grey, and the compartments for counter flooding with light blue. The first water volume values are those in the numerical model, and the values in brackets are the values realized in the scale model used in the model tests.

In this case, a rapid counter flooding using an opening size of 1.5 m² to sea for each counter flooding compartment is sufficient to prevent ship capsize, which succeeds, and the ship survives 1800 s easily. The development of the roll angle as a function of time together with the water volumes on the trailer deck and in the compartments are shown in Figure 6 for both cases, with and without counter flooding. The solid red curve shows the roll angle without counter flooding, and the dash-dotted one shows the roll angle with counter flooding. The bluish curves show the floodwater volumes due to the damage opening, and the two green ones due to counter flooding.

In the second test case, the damage opening time of 30 s was used. The counter flooding was assumed to start when the ship heeling angle exceeds 5° as before. As the damage opening time is somewhat longer, it is sufficient to use an opening size of 1.1 m² for each counter flooding compartment, not only to prevent a rapid ship capsize, but also a later loss due to gradual flooding. The mitigation succeeds for both cases and the ship survives 1800 s easily, as shown in Figure 7.

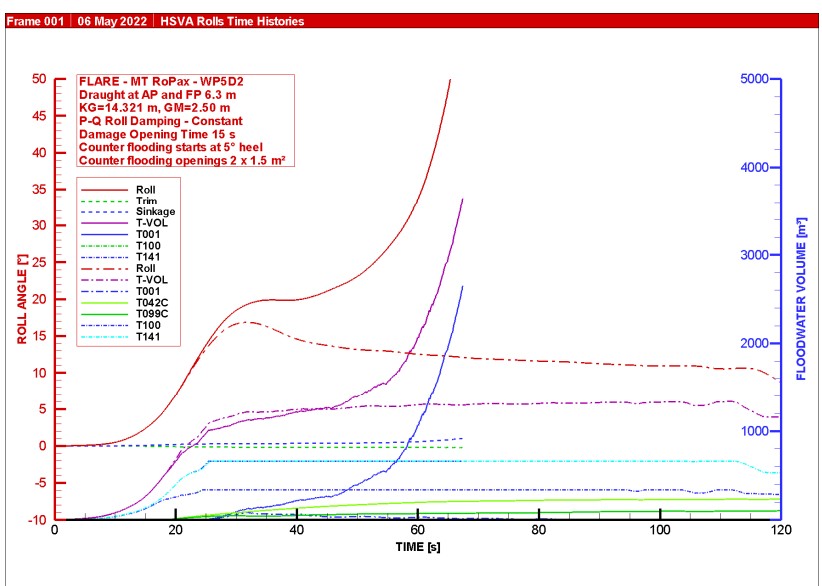

**Figure 6.** Counter flooding with damage opening time 15 s: The roll angle (red) and water volumes on trailer deck and in damaged compartments (bluish) and in those for counter flooding (green) are shown. T-VOL is the total volume of the floodwater in ship, and T001 is the water volume on the trailer deck.

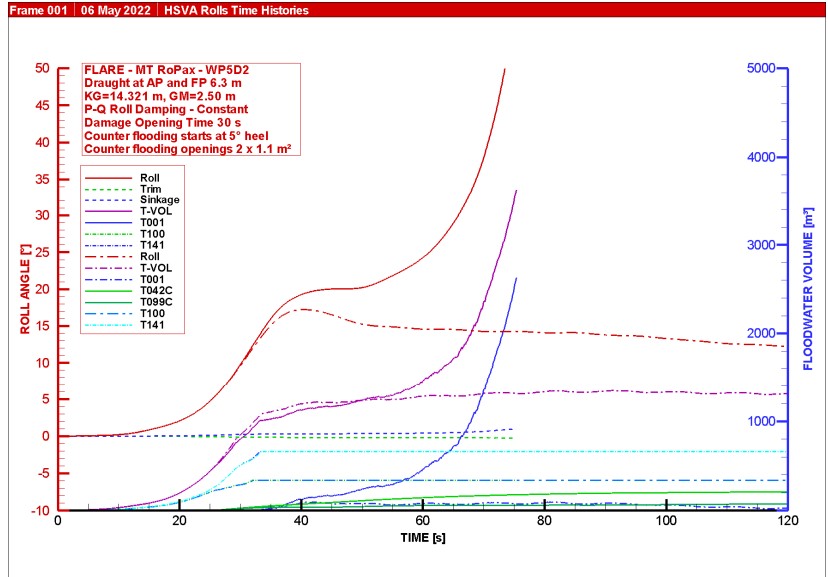

**Figure 7.** Counter flooding with damage opening time 30 s: The roll angle (red) and water volumes on trailer deck and in damaged compartments (bluish) and in those for counter flooding (green) are shown.

The computations show that even with the low GM value of 2.5 m, the counter flooding prevents the rapid capsize of the ship. With the applied relatively short damage opening times, the compartments used for counter flooding need to be filled rapidly. This requires duct opening sizes for counter flooding in these compartments of about 1.1 m² to 1.5 m². As further study of the water intake into these compartments is beyond the scope of the present investigation, each of them was opened to sea at ship side 1.0 m in height from the compartment bottom, and 1.1–1.5 m in length for the studied cases, respectively. This leads to duct opening sizes of 1.1 m² or 1.5 m² for each counter flooding compartment. The damage case was further studied with the lowest ship GM 3.4 m on the limiting curve at a ship draught of 6.3 m, with the damage opening time of 15 s, in beam seas with significant wave heights Hs (2.0), 2.5, 3.0, 3.5, and 4.0 m. The results are shown below in Table 2.

**Table 2.** Damage Case 2, with (red) and without (black) counter flooding in beam seas. The values in brackets were not explicitly computed, as the result (survival) is clear.

| $H_S$ [m] | \multicolumn{10}{c}{DC2 with GM 3.4 m, DOT 15 s, counter flooding start at 5°, Heel—TTC shown in [s], 1800 = survival. Duct openings from sea to CF compartments = 2 × 1.5 m², $T_p$ = 10.0 s} |
|---|---|---|---|---|---|---|---|---|---|---|
| $H_S$ [m] | 2.0 | 2.0 | 2.5 | 2.5 | 3.0 | 3.0 | 3.5 | 3.5 | 4.0 | 4.0 |
| Mitigation | No | Yes | No | Yes | No | Yes | No | Yes | No | Yes |
| 1 | 1800 | (1800) | 1800 | 1800 | 496.3 | 1800 | 412.2 | 1800 | 145.9 | 528.7 |
| 2 | 1800 | (1800) | 1800 | 1800 | 230.3 | 1800 | 122.1 | 1800 | 92.1 | 249.9 |
| 3 | 1800 | (1800) | 1150 | 1800 | 410.1 | 1800 | 328.6 | 1800 | 203.9 | 431 |
| 4 | 1800 | (1800) | 459.8 | 1800 | 299.1 | 1800 | 116.6 | 1800 | 80 | 132.8 |
| 5 | 1800 | (1800) | 252.4 | 1800 | 67.5 | 192 | 58.2 | 70.2 | 50.3 | 54.2 |
| 6 | 1800 | (1800) | 466.5 | 1800 | 216.7 | 514.6 | 159.3 | 467.7 | 139.5 | 227.2 |
| 7 | 1800 | (1800) | 356.7 | 1800 | 312 | 1800 | 247.8 | 369 | 222.9 | 279.6 |
| 8 | 1800 | (1800) | 1800 | 1800 | 1800 | 1800 | 380.7 | 1800 | 245.4 | 1800 |
| 9 | 1800 | (1800) | 1800 | 1800 | 202.5 | 1800 | 120.2 | 964.3 | 114.7 | 155.2 |
| 10 | 1800 | (1800) | 1800 | 1800 | 228.4 | 1800 | 89.5 | 1800 | 71.5 | 109.6 |
| Survival | 10/10 | (10/10) | 5/10 | 10/10 | 1/10 | 8/10 | 0/10 | 6/10 | 0/10 | 1/10 |

Table 2 shows the computed times to capsize with and without counter flooding for 10 irregular wave sequence realizations. The following observations can be made:

- At the low significant wave height $H_S$ 2.0 m, there is no acute need for counter flooding as the ship survives also without. However, if counter flooding is used, this results in a lower heeling angle, which would be beneficial for all rescue and disembarkation operations on the ship;
- At significant wave heights $H_S$ 2.5, 3.0, and 3.5 m, the counter flooding results in a significant improvement in the ship survivability. The corresponding survival rates increase from 50% (%) to 100%, 10% to 80%, and 0% to 60%, respectively;
- At significant wave height $H_S$ 4.0 m, the counter flooding slightly improves the time to capsize, but in this higher sea state the ship is able to survive due to gradual flooding only in 10% of the cases. Thus, the effect of flooding mitigation is small. The capsize mechanism in higher sea states is, as usual in RoPax ships, the further accumulation of water on the trailer deck, even if in this case the extent of the trailer deck is already limited with transverse bulkheads;
- It is noteworthy that the mitigation in Damage Case 2 considerably improves the ship survivability in a very large portion of the most common sea states, being very effective at the wave heights $H_S$ 2.5–3.5 m, the range of which covers a large portion of those sea states at which ship-to-ship collisions statistically tend to take place. Thus, the mitigation through counter flooding is effective in the relevant, most common sea state range. A duct opening size for each counter flooding compartment of about 1.1 m² to 1.5 m² is required.

*4.4. Model Test Results on Damage Case 2 with and without Counter Flooding*

In Damage Case 2 (DC2), the compartments T100 (max. volume 327 m³) and T141 (708 m³) on the starboard side of the MT RoPax get damaged and are flooded, as illustrated in Figure 5. The damage and its opening extensions are identical to those in the numerical model. Due to technical reasons, the floodable compartment volumes in the ship model are somewhat smaller than in the numerical model. It was necessary to use a higher GM value in the model tests than in the preceding numerical computations.

Without mitigation, the ship with GM 3.018 m capsizes in calm water in about 65–85 s when the damage opens in 30 s. With a damage opening time of 15 s, these capsize times would be about 10 s shorter. With GM 3.48 m, the ship capsizes in beam seas with $H_S$ 3.5

m in ca. 80% of the cases when the damage is opened in 15 s. With $H_s$ 5.0 m, the ship capsizes in ca. 100% of the cases when the damage is opened in 15 s.

In the experiments, no valve can open in zero-time duration and, in addition, the two compartments were flooded through inflow pipes of considerable length (ee Appendix A). The inertia of the water mass in these pipes delays the start of flooding in comparison with the computations. For this reason, an earlier starting point for the counter flooding with 1° roll angle was necessary. The inside diameter of the mentioned pipes was 49 mm in the model tests, which leads to a 1.5 m² cross-sectional area in f.sc. for the flow in the pipe, which is the same as in the numerical model.

Figure 8 shows the experimental development of the roll angle as a function of time upon damage opening in 30 s in calm water with and without counter flooding. In all six cases without counter flooding, the ship capsized rapidly, whereas the counter flooding prevented capsize in all five cases. The horizontal red line at 1° shows when the valves to flood the two counter flooding compartments open automatically.

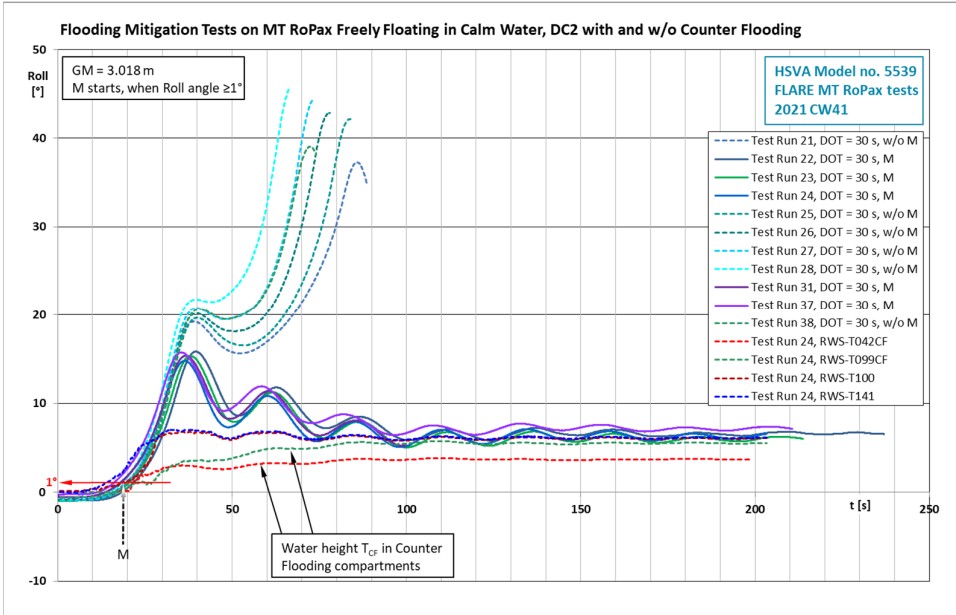

**Figure 8.** Damage Case 2 with damage opening time 30 s: Roll angle as a function of time with (solid lines) and without (dashed lines) counter flooding. The dashed curves below show the water height in the damaged compartments T100 and T141 and also in those for counter flooding T042CF and T099CF for Test Run 24.

Figure 9 shows the corresponding experimental development of the roll angle with a damage opening time of 15 s in calm water with counter flooding only. With the shorter damage opening time of 15 s, the counter flooding was in one case out of six not fast enough to stabilize the ship and prevent capsize. This yields a survival rate with counter flooding of 83%. Without mitigation, the ship would capsize rapidly in all cases, leading to a survival rate of 0%.

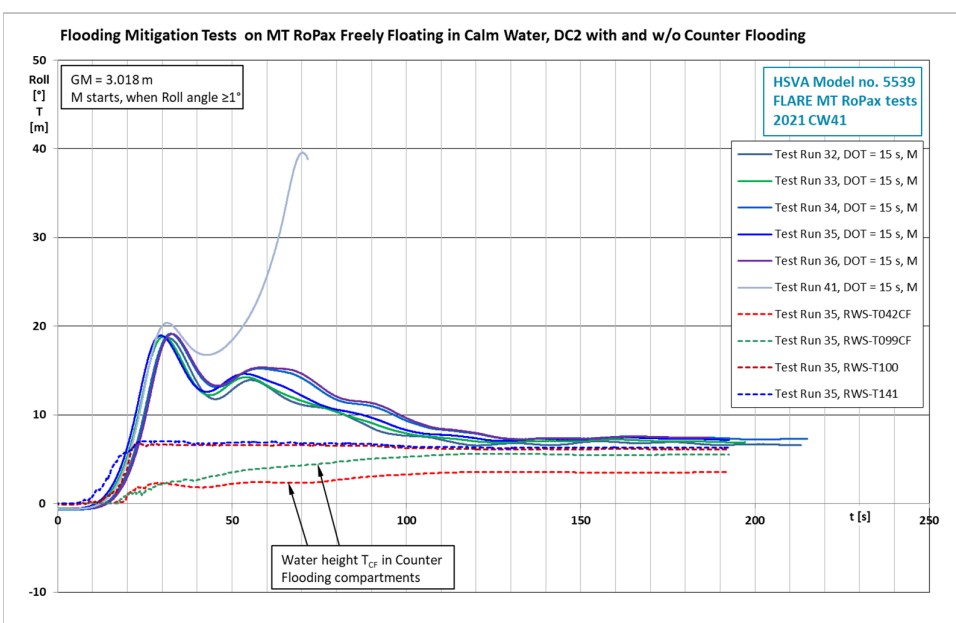

**Figure 9.** Damage Case 2 with damage opening time 15 s: Roll angle as a function of time with counter flooding (solid lines). The dashed curves below show the water height in the damaged compartments T100 and T141 and also in those for counter flooding T099CF and T042CF for Test Run 35.

With GM 3.48 m, the ship survived in beam seas of significant wave height $H_S$ 3.5 m in 100% of the cases when the damage was opened in 15 s. Without mitigation, the survival rate was only 20%.

With $H_S$ 5.0 m, the ship survived in ca. 100% of the cases when the damage was opened in 15 s. Without mitigation, the ship capsized in all cases.

Due to the counter flooding, the heeling angle after the transient phase stays at ca. 7°. This value should pose no serious problems for evacuation or other activities onboard. Even though the ship draught has increased through the floodwater and also due to the counter flooding, the reduction of the remaining heeling angle is more significant and the ship has a good survivability in waves after the mitigation.

With the applied relatively short damage opening times of 15 s and 30 s, the compartments used for counter flooding need to be filled rapidly. Based on preliminary numerical simulations, a duct opening size for counter flooding in these compartments of 1.5 m² was used in the model tests. The longer than anticipated inflow pipes in the ship model somewhat delayed the inflow and for this reason an earlier counter flooding start in the experiments needed to be used. In a real ship in full scale, the ducts can, due to constructional reasons, be shorter, thus allowing a faster initial flow rate and thus a later counter flooding start. Altogether, the counter flooding is in the studied case a very successful flooding mitigation method, as shown by the experimental model test results.

*4.5. Numerical Hindcast of the Flooding Mitigation with Counter Flooding*

The a priori numerical simulations were used to define the parameters for the model tests. As some technical details in the physical ship scale model deviated from those in the preceding simulations, a hindcast with a more accurate and improved numerical simulation was carried out. The compartment volumes and inflow duct lengths were accurately set to those in the physical model and improved numerical modeling was used: a dynamic orifice equation was used for compartment inflow instead of the steady-state Bernoulli model [9]. This was particularly important for the counter flooding compartments with the long inflow ducts. The transient flow into the large open compartment T141 was modeled with shallow-water-equations instead of the usual pendulum model. These measures

improved the accuracy of the numerical modeling. However, some differences between the numerical simulations and model tests results remain. Figure 10 illustrates the situation. In general, such remaining differences are mainly due to shortcomings in the numerical modeling and much less due to scale effects in the model tests.

The correlation between the model test results and the numerical simulation in Figure 10 is altogether very satisfactory. As seen in the pronounced oscillations in all experimental curves, there are more water dynamics present in the experiments than in the numerical results. However, the experimental and numerical results were achieved with slightly different GM values. The in inclination tests measured GM values are reliable when measured, but in the course of several mitigation tests, some rest water may accumulate in the ship model hull, having a reducing effect on the actual GM of the vessel. Thus, the given experimental GM values may be slightly too high to be used directly in computations.

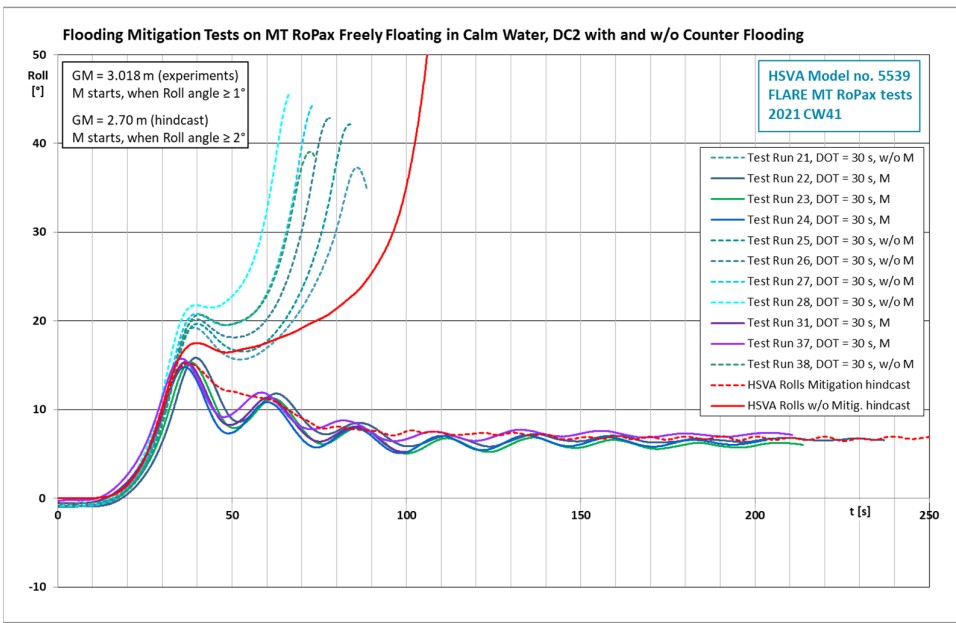

**Figure 10.** Damage Case 2 with damage opening time 30 s: Roll angle as a function of time with (solid lines) and without (dashed lines) counter flooding in model tests. The red solid line shows the numerical hindcast without mitigation, the dashed red line with mitigation.

### 4.6. Conclusions on Damage Case 2

- The mitigation computations on Damage Case 2 show that the counter flooding in calm water and in lower-to-middle sea states can prevent the ship from capsizing, and can thus have a significant effect on the potential loss of life (PLL);
- In higher sea states in beam seas, the counter flooding prevents the ship from rapid capsize, but the ship with the damaged and counter flooding compartments being all flooded has a reduced survivability. Thus, when the significant wave height increases to $H_S$ 4.0 m, the ship with GM 3.4 m starts, according to simulations, to capsize. However, even in these cases, capsize is delayed, allowing more passengers and crew to disembark the ship, reducing the potential loss of life (PLL). According to the model tests, the damaged ship with intact GM 3.48 m still survives $H_S$ 5.0 m in 100% of the cases with mitigation;
- The counter flooding has the significant advantage that few best suitable compartments on each side of the ship can be prepared for this. Their ship stabilizing effect can be applied to a large variety of damage cases on the opposite side of the ship. In the particular case of the MT RoPax, it is not difficult to find two to four suitable compartments, for example void spaces, on each ship side for this purpose;

- The present numerical modeling techniques available are sufficient for quite-accurate numerical modeling of ship behavior with counter flooding when attention is paid to the dynamic character of the flooding phenomena.

### 4.7. Numerical Simulation of Damage Case 4 with and without Recovery of Lost Buoyancy

In Damage Case 4 (MSRC DMC0385; DC4), the trailer deck T001, and the compartments below T043 (max. volume 586 m³), T045 (714 m³), T046 (776 m³), T083 (303 m³), and T098 (262 m³) on the starboard side get damaged and are flooded (see Figure 11). The damage opening size is about 8.19 m in length and extends 4.01 m below the still water level at the ship draught of 6.3 m. Without mitigation, the ship capsizes with a GM value of 2.3 m in about 150 s when the damage opens in 15 s. This is near the limit of capsizing or surviving, as the vessel survives with GM 2.4 m. With the original design GM 3.40 m, the ship does not capsize in calm water or beam seas with a damage opening time of 15 s until the significant wave $H_s$ 5.0 m is reached. Thus, the mitigation efforts in calm water were studied using the lowest GM value 2.3 m and damage opening time 15 s, in which conditions the ship capsizes without mitigation.

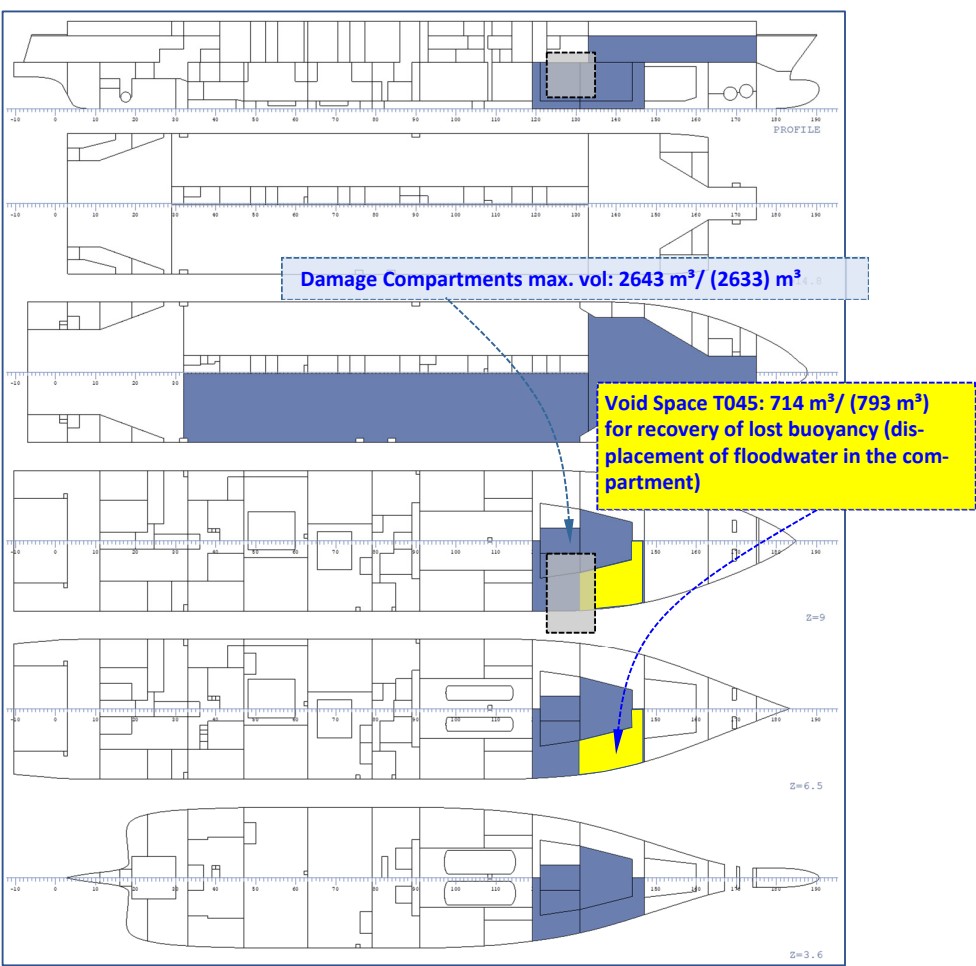

**Figure 11.** Damage Case 4 with recovery of lost buoyancy: The grey rectangle shows the damage penetration. The damaged compartments T001, T043, T046, T083, T098, and T045 are shown with dark blue-grey or yellow, the T045 with yellow being the compartment for recovery of lost buoyancy through floodwater displacement. The first water volume values are those in the numerical model, the values in brackets are the values realized in the scale model used in the model tests.

The mitigation effort consisted of displacing floodwater in the compartment T045 on the damaged starboard side of the ship, or in other words, recovering lost buoyancy in it (see the yellow compartment in Figure 11). As in the previous damage case, the damage opening time 15 s was used.

In the numerical simulations, the floodwater in the compartment T045 (714 m³) was displaced by linearly reducing the permeability in the compartment to model the filling of the compartment with a substance (e.g., foam) having a specific density of 50 kg/m³. The floodwater in the compartment was displaced starting 60 s after the end of the damage opening—75 s from the start of the damage opening—and ending after 180 s—255 s after the start of the damage opening.

The mitigation is sufficient to prevent capsize and the ship survives 1800 s easily, as shown in Figure 12. The solid curves show the case without mitigation, the dashed ones with. The damage case was further studied with the lowest ship GM value 3.4 m on the limiting curve at the ship draught of 6.3 m, with the damage opening time 15 s, in beam seas with significant wave heights $H_S$ (2.0), 5.0, 5.5, 6.0, and 7.0 m. The results are shown in Table 3.

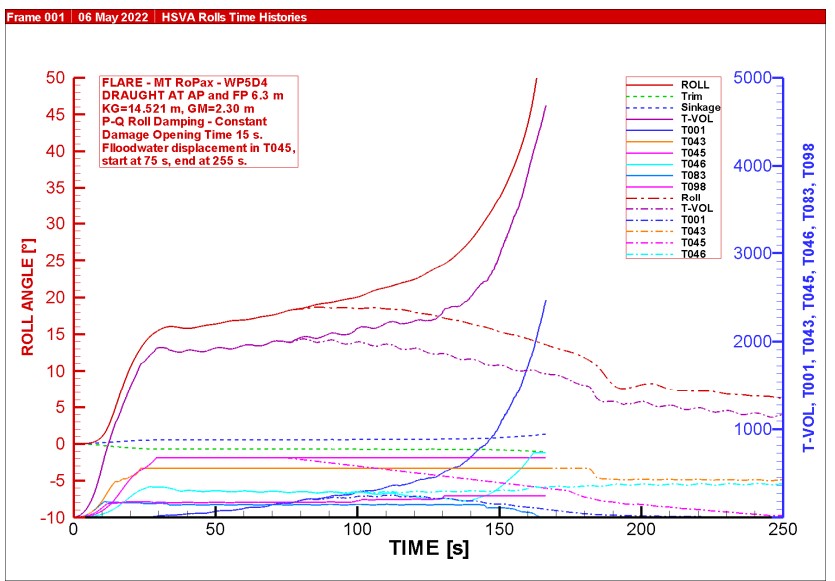

**Figure 12.** Recovery of lost buoyancy with damage opening time 15 s: The roll angle (red) and water volumes on trailer deck (blue) and in damaged compartments (bluish, reddish, deep yellow) and in the compartment T045 (pink solid/dash-dotted curves) chosen for floodwater displacement. T-VOL is the total volume of the flood water in ship, T001 the water volume on the trailer deck.

**Table 3.** Damage Case 4, with (red) and without (black) recovering the lost buoyancy in compartment T045, in beam seas. The values in brackets were not explicitly computed, as the result (survival) is clear.

| DC4 with GM 3.4 m, DOT 15 s, RLB: displacement of floodwater in T045 starts at 75 s, ends at 255 s, —TTC shown in [s], 1800 = survival, $T_p$ = 10.0 s | | | | | | | | | |
|---|---|---|---|---|---|---|---|---|---|
| $H_S$ [m] | 2.0 | 2.0 | 5.0 | 5.0 | 5.5 | 5.5 | 6.0 | 6.0 | 7.0 | 7.0 |
| Mitigation | No | Yes | No | Yes | No | Yes | No | Yes | No | Yes |
| 1 | 1800 | (1800) | 1800 | 1800 | 1132.6 | 1800 | 843.0 | 1800 | 430.8 | 492.9 |
| 2 | 1800 | (1800) | 1800 | 1800 | 1800 | 1800 | 636.8 | 1800 | 468.6 | 190.0 |
| 3 | 1800 | (1800) | 1800 | 1800 | 706.3 | 1800 | 707.2 | 1800 | 418.8 | 387.8 |
| 4 | 1800 | (1800) | 1800 | 1800 | 1800 | 1800 | 495.6 | 1800 | 218.1 | 249.9 |
| 5 | 1800 | (1800) | 1800 | 1800 | 1800 | 1800 | 1800 | 1800 | 349.7 | 155.9 |
| 6 | 1800 | (1800) | 611.2 | 1800 | 426.4 | 548.5 | 303.6 | 493.1 | 178.1 | 179.4 |
| 7 | 1800 | (1800) | 1187.7 | 1800 | 348.7 | 349.1 | 347.9 | 332.1 | 336.6 | 280.1 |
| 8 | 1800 | (1800) | 1800 | 1800 | 1800 | 1800 | 814.2 | 1800 | 641.3 | 687.5 |
| 9 | 1800 | (1800) | 1800 | 1800 | 890.9 | 1800 | 300.0 | 223.3 | 184.9 | 170.1 |
| 10 | 1800 | (1800) | 1800 | 1800 | 841.5 | 1800 | 842.0 | 1800 | 215.2 | 100.7 |
| Survival | 10/10 | (10/10) | 8/10 | 10/10 | 4/10 | 8/10 | 1/10 | 7/10 | 0/10 | 0/10 |

The table shows the computed times to capsize with and without the recovery of lost buoyancy in the flooded compartment T045 for 10 irregular wave sequence realizations.

The following observations can be made:

- At low significant wave heights $H_s$ 2.0 m–4.5 m there is no acute need for recovering the lost buoyancy in compartment T045, as the ship survives also without. However, if the mitigation method is used, this results in a lower heeling angle, which would be beneficial for all rescue and disembarkation operations on the ship;

- At significant wave heights $H_s$ 5.0, 5.5 m, and 6.0 m, recovering the lost buoyancy results in a clear and significant improvement of the ship survivability. The corresponding survival rates increase from 80% to 100%, 40% to 80%, and 10% to 70%, respectively;

- At significant wave height $H_s$ 7.0 m, recovering the lost buoyancy in compartment T045 does not improve the time to capsize, as in this higher sea state the ship is not able to survive the gradual flooding. Thus, the effect of mitigation is practically non-existent. The capsize mechanism in higher sea states is, as usual in RoPax ships, the further accumulation of water on the trailer deck, even if the extent of the trailer deck is already limited, as shown in Figure 11;

- The mitigation in Damage Case 4 with the recovery of the lost buoyancy in compartment T045 considerably improves the ship survivability in a large portion of the prevailing sea states, being very effective at the wave heights $H_s$ 5.0–6.0 m, the range of which covers a portion of the prevailing sea states. In Damage Case 4, the ship survives well in lower beam sea states without mitigation. If the damage were larger, the effective range of mitigation would be located at lower, more frequent sea states. The mitigation through the recovery of lost buoyancy in a damaged compartment is a suitable method for this;

- In its simplest form, the recovery of lost buoyancy in a damaged compartment reduces the heeling angle of the ship and thus in most cases also the water ingress on the trailer deck, which is crucial for RoPax ship survival. The reducing effect of the mitigation on ship draught is likely to be less important.

*4.8. Model Test Results on Damage Case 4 with and without Recovery of Lost Buoyancy*

In Damage Case 4, the trailer deck T001, and the compartments below T043 (max. volume 515 m³), T045 (793 m³), T046 (735 m³), T083 (383 m³), and T098 (207 m³) on the starboard side get damaged and are flooded (see Figure 11). The damage and its opening extensions are identical to those in the numerical model. Due to model construction, the floodable compartment volumes in the ship model show some deviations from those in the numerical model. It was also necessary to use a higher GM value in the model tests than in the preceding numerical computations: without mitigation, the ship capsizes with a GM value of 3.71 m in about 150–180 s, when the damage opens in 15 s. Thus, the mitigation efforts in calm water were studied using this GM value.

The mitigation effort consisted of displacing floodwater in the compartment T045 (void space) on the damaged starboard side of the ship (see the yellow compartment in Figure 11). As in the previous case, the damage opening time 15 s was used. In the a priori numerical simulations, the floodwater in the compartment T045 (714 m³ in the numerical model) was displaced by linearly reducing the permeability in the compartment. For technical reasons in the model tests, it was possible to displace the f.sc. floodwater volume of only 483 m³ out of the total compartment volume of 793 m³ of the compartment T045. For this reason, the reaction time (RT) to start the mitigation (M) in 60 s based on the a priori simulations could not be used, but a shorter reaction time of 45 s needed to be used. Thus, the mitigation was started 45 s after the damage opening in 15 s was completed; that is, 60 s after the start of the damage opening. The set-point filling time of the inflatable container with pressurized air was kept unchanged at 180 s, but the effective duration of the floodwater displacement in the model tests turned out to be much shorter, ca. 70 s. This value is based on video analysis after the tests (see Figure 13).

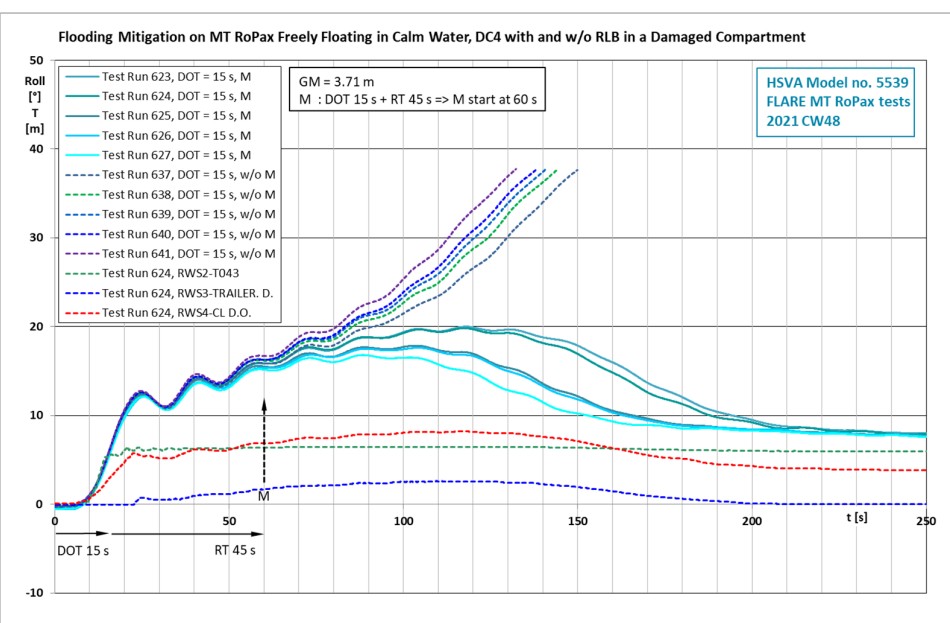

**Figure 13.** Damage Case 4 with damage opening time 15 s: Roll angle as a function of time with (solid lines) and without (dashed lines) recovery of lost buoyancy in the damaged compartment T045. The dashed curves below show the water height in the damaged compartments T043, on the trailer deck sensor, and just outside of the damage opening at its centerline for Test Run 624.

With the damage opening time of 30 s, the reaction time needed to be further shortened to 30 s. Thus, the mitigation was started 30 s after the damage had opened in 30 s; that is, also in this case 60 s after the start of the damage opening (see Figure 14).

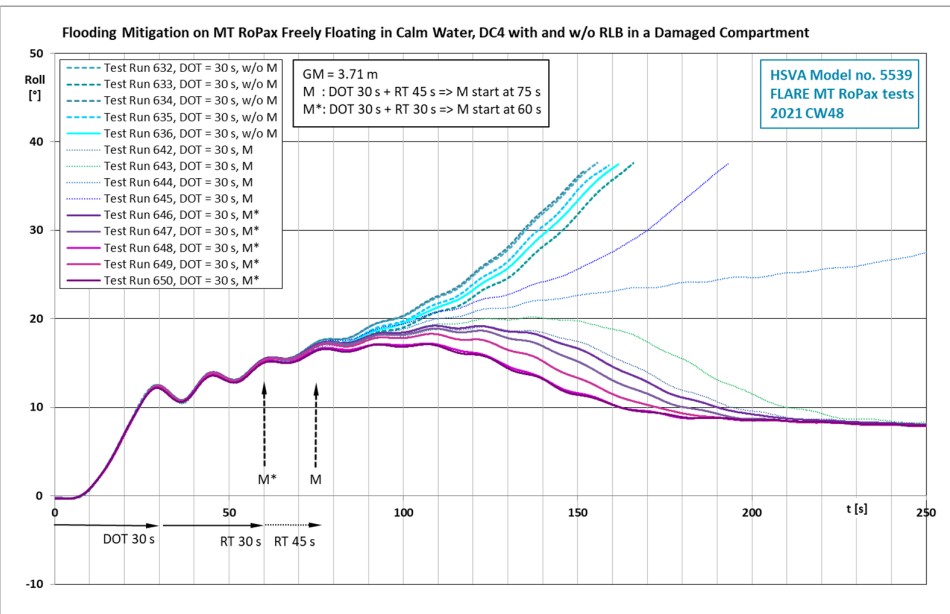

**Figure 14.** Damage Case 4 with damage opening time 30 s: Roll angle as a function of time with (solid lines) and without (dashed lines) recovery of lost buoyancy in the damaged compartment T045. The dotted curves show cases ('M') in which the reaction time 45 s did not in all cases lead to ship survival.

The ship with GM 3.71 m survived in calm water in all test cases with both damage opening times 15 s and 30 s; that is, 100% of the cases. Without mitigation, the ship capsized rapidly in all cases.

With GM 3.81 m, the ship survived with flooding mitigation in beam seas with $H_S$ 2.5 m in 100% of the cases when the damage was opened in 15 s, with the mitigation reaction time being 45 s. Without mitigation, the ship capsized in all cases. With the significant wave height $H_S$ 4.0 m, the ship survived with flooding mitigation in 80% of the cases. Without mitigation, it capsized in all cases.

### 4.9. Numerical Hindcast of the Flooding Mitigation with Recovery of Lost Buoyancy

The a priori numerical simulations were used to define the parameters for the model tests. Additionally, in this case, a hindcast with a more accurate and improved numerical simulation was carried out. The compartment volumes, as in the physical ship model, and the actual starting time and achieved duration of the filling of the inflatable container with pressurized air to displace floodwater, were used in the hindcast. This considerably improved the accuracy of the numerical hindcast. The use of the dynamic orifice equation instead of Bernoulli's equation for the inflow was not relevant in this case. All compartments below the trailer deck were modeled with the pendulum model, which is suitable for simple modeling of the complicated compartment geometries at hand. As seen in the oscillations in all experimental curves in Figure 15, there are more water dynamics present in the experiments than in the smoother numerical results. Altogether, the correlation between the model test results and the numerical simulation in Figure 15 is very satisfactory. However, the experimental and numerical results were achieved with different GM values. The difference is too large to be explained with the small inaccuracies in GM during the course of several individual model test runs. The difference in the model tests and the hindcast is more attributable to the limited ability of the numerical model to account for the rapid flooding of very complicated compartments in the damage area. Due to the complicated shapes of the compartments, modeling with shallow-water-equations was not possible, but a simple pendulum model needed to be used, which does not always provide sufficient modeling in very dynamic cases.

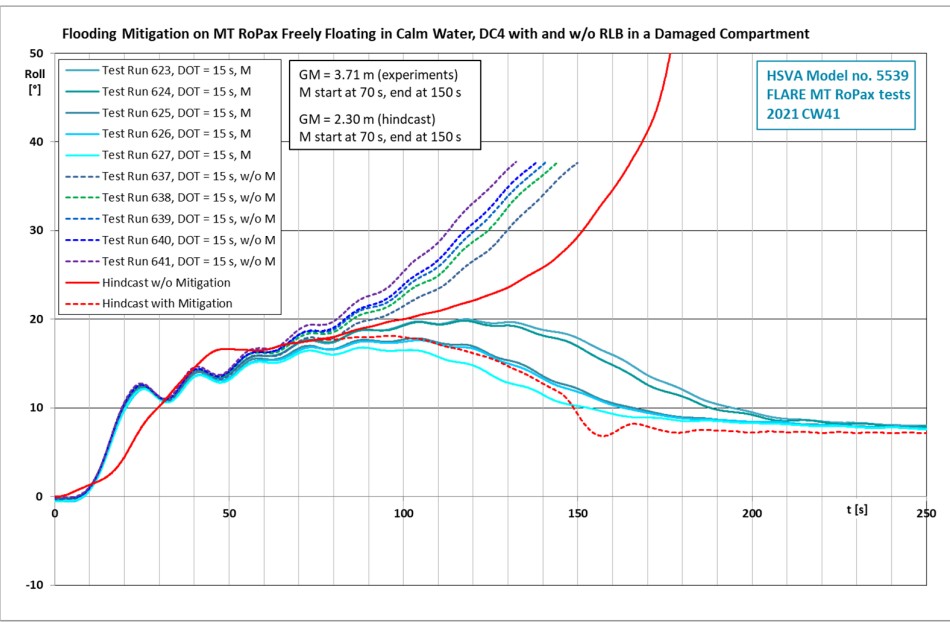

**Figure 15.** Damage Case 4 with damage opening time 15 s: Roll angle as a function of time with (solid lines) and without (dashed lines) recovery of lost buoyancy in the damaged compartment T045. The red solid line shows the numerical hindcast without mitigation, the dashed red line the hindcast with mitigation.

*4.10. Conclusions on Damage Case 4 with and without Recovery of Lost Buoyancy*

- The flooding mitigation with the recovery of lost buoyancy in a damaged compartment is sufficient to prevent capsize and the ship survives 1800 s easily in the studied cases;
- In its simplest form, recovering lost buoyancy in a damaged compartment reduces the heeling angle of the ship and thus in most cases also water ingress on the trailer deck, which is crucial for RoPax ship survival. The reducing effect of the mitigation on the draught is likely to be less important;
- The righting lever provided by the recovery of the lost buoyancy in a damaged compartment is an important factor for the effectiveness of the mitigation. This depends on the compartment volume and the distance of its center of volume to the centerline of the ship;
- Recovering lost buoyancy in the damaged compartments in principle implies the preparation of each compartment that can get damaged for this. At least preparation of the largest compartments far away from the centerline is necessary;
- The idea to displace flood water in a damaged compartment with a lighter substance, e.g., expandable foam, will always leave some questions open with respect to the functionality of such a mitigation system in a compartment possibly already heavily damaged by a collision or grounding;
- On one hand, the flooding mitigation with the recovery of lost buoyancy is more difficult to arrange than counter flooding. On the other hand, some of the buoyancy lost in the damaged compartment is recovered, which improves the ship survivability;
- As the recovery system of the lost buoyancy in a damaged compartment system should be available at any arbitrary compartment that can get damaged, it is difficult to consider that the system could be deployed very rapidly. In this study, the reaction times needed to be shortened from those originally planned to yield positive results. During the chosen mitigation time of 180 s between ca. 500 $m^3$ and 793 $m^3$, floodwater needed to displaced out of the damaged compartment. This is a formidable task.

*4.11. Numerical Simulation of Damage Case 4 with and without Deployment of a Watertight Barrier on the Trailer Deck*

A further test of a mitigation effort with a deployable watertight barrier (WTB) against floodwater spreading on the trailer deck of the MT RoPax was modeled in the numerical simulation by blocking the flow in the longitudinal direction on the trailer deck at the building frame # 81, 64.8 m in front of the AP (see Figure 16). The barrier is deployed with a delay of 30 s, describing the deployment time needed to lower the curtain barrier in a ship. This time is based on the descending times of several rolling gates presently in use and is suitable for a truck passage underneath. After this time, the barrier is assumed to be watertight in the low water heights on the trailer deck, even if some further filling or reinforcement of the barrier would still be going on. The barrier deployment starts as soon as the water depth of 0.07 m is reached at a chosen location along the outer starboard-side edge of the trailer deck between the barrier and the damage location. In the numerical simulation, this takes place as soon as the water height in the chosen corresponding grid cell of the numerical grid for the shallow-water-equations exceeds this threshold value.

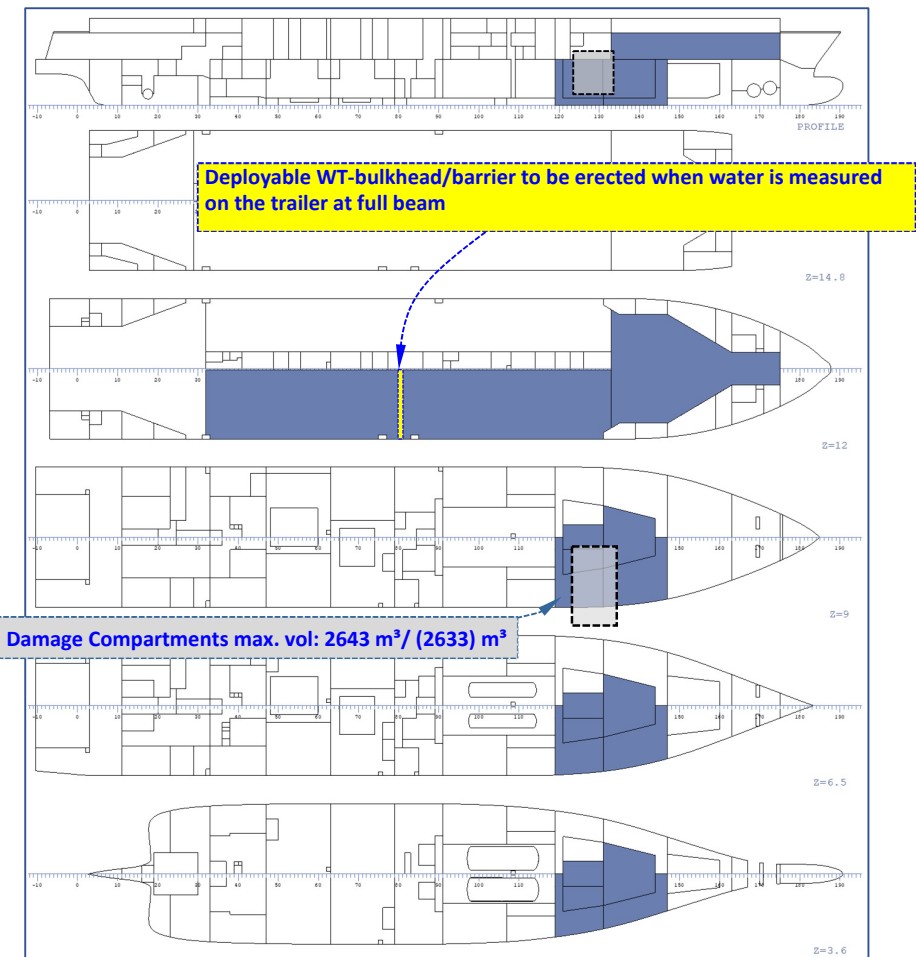

**Figure 16.** Damage Case 4 with watertight barrier on the trailer deck: The grey rectangle shows the damage penetration. The damaged compartments T001, T043, T045, T046, T083, and T098 are shown with dark blue-grey, the barrier to block the floodwater spreading on the trailer deck with yellow. The first water volume is the one in the numerical model, the value in brackets is the value realized in the scale model used in the model tests.

The modeled release/trigger system for lowering the curtain barrier works properly and the system is able to prevent the ship capsize in the simulation, as illustrated in Figure 17. The curtain barrier limits the horizontal floodwater extent and thus also its volume on the trailer deck. The flooding of the other damaged compartments is not mitigated. Therefore, the mitigation does not recover anything. This results in a significant remaining heeling angle, which can slow down any evacuation effort on the damaged ship, would such an effort become necessary. As with the case of recovery of lost buoyancy, the mitigation efforts in calm water were simulated using the GM value 2.3 m, and a damage opening time of 15 s, in which conditions the ship capsizes without mitigation.

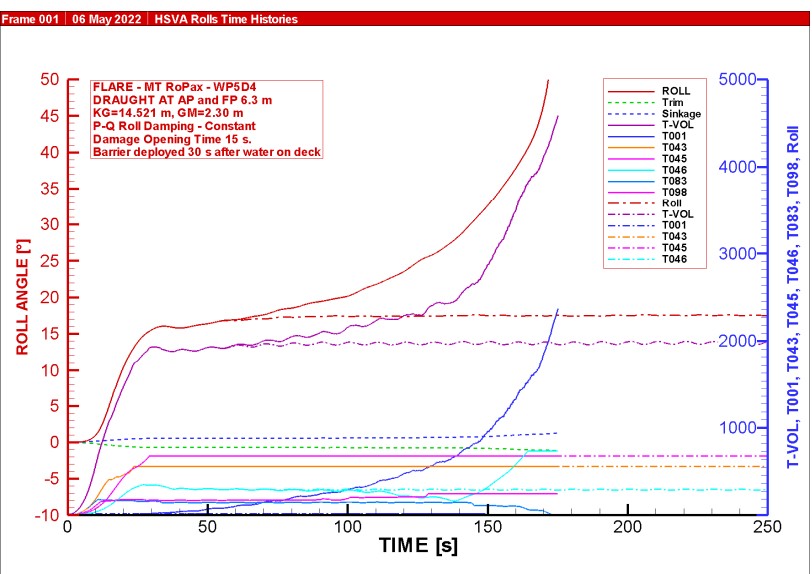

**Figure 17.** Deployable barrier (in 30 s) with damage opening time 15 s: The roll angle (red) and water volumes on trailer deck (deep blue) and in damaged compartments. The solid curves show the situation without the deployed barrier, the dash-dotted ones the results with the barrier.

The mitigation effort was further studied for the ship with GM 3.4 m in irregular beam sea states with $H_S$ 5.0, 5.5, 6.0, and 7.0 m. Additionally, at the higher sea states, the ship with mitigation is able to show a high survivability. The mitigation effort not only prevents a rapid capsize, but also compensates the effects of flooding to a sufficient degree, and the damaged ship is able to survive in higher sea states.

The unavoidable delay in the automatic lowering of the curtain barrier poses a potential problem possibly requiring some attention: During the damage opening, and later during the deployment time, before the curtain barrier locks into its final position, floodwater can flow onto the trailer deck, also to areas on the deck further beyond the barrier, as shown by the simulation in Figure 18. After the floodwater flows past the barrier, its way back is prevented once the barrier is down. The following improvements would be useful:

- The deployment time of such a barrier should be as short as possible;
- If any floodwater goes past the barrier, there should a possibility to let this water to flow out.

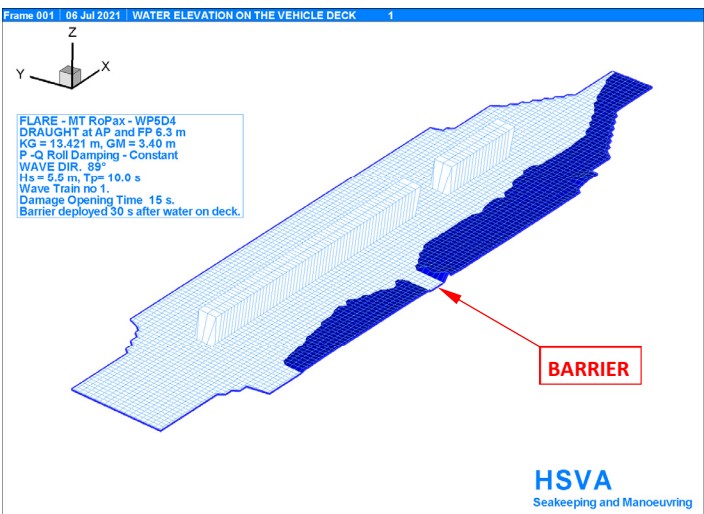

**Figure 18.** Water on the trailer deck on both sides of the closed barrier preventing further spreading of the floodwater.

Table 4 shows the computed times to capsize with and without deployment of the WT barrier for the real ship GM value of 3.40 m for the draught 6.3 m. The following observations can be made:

- At significant wave heights up to *H*s 4.0 m, there is no acute need for the deployment of the WT barrier in the given damage case, as the ship survives with the lowest GM value 3.4 m on the limit curve also without;
- At significant wave heights *H*s 5.0, 5.5, 6.0, and 7.0 m, the deployment of the WT barrier results in a clear and significant improvement of the ship survivability. With the ship GM value 3.4 m, the corresponding survival rates increase from 80% to 100%, from 40% to 100%, from 10% to 100%, and from 0% to 100%, respectively;
- The mitigation in Damage Case 4 with the deployment of a WT barrier considerably improves the ship survivability in a large portion of the prevailing sea states, being very effective at the wave heights *H*s 5.0–7.0 m. If the damage were larger, the effective range of mitigation would be located at lower, more frequent wave heights. The mitigation through the deployment of the WT barrier is a suitable method for this.

**Table 4.** Damage Case 4 with (red) and without (black) deploying the WT barrier on the trailer deck in beam seas.

| DC4 with GM 3.4 m, DOT 15 s, WTB Deployment time 30 s, TTC shown in [s], 1800 = survival, *T*p = 10.0 s | | | | | | | | |
|---|---|---|---|---|---|---|---|---|
| *H*s [m] | 5.0 | 5.0 | 5.5 | 5.5 | 6.0 | 6.0 | 7.0 | 7.0 |
| Mitigation | No | Yes | No | Yes | No | Yes | No | Yes |
| 1 | 1800 | 1800 | 1132.6 | 1800 | 843.1 | 1800 | 430.9 | 1800 |
| 2 | 1800 | 1800 | 1800 | 1800 | 636.9 | 1800 | 468.7 | 1800 |
| 3 | 1800 | 1800 | 706.3 | 1800 | 707.3 | 1800 | 418.9 | 1800 |
| 4 | 1800 | 1800 | 1800 | 1800 | 495.7 | 1800 | 218.2 | 1800 |
| 5 | 1800 | 1800 | 1800 | 1800 | 1800 | 1800 | 349.8 | 1800 |
| 6 | 611.3 | 1800 | 426.4 | 1800 | 303.7 | 1800 | 178.2 | 1800 |
| 7 | 1187.8 | 1800 | 348.7 | 1800 | 348.0 | 1800 | 336.7 | 1800 |
| 8 | 1800 | 1800 | 1800 | 1800 | 814.3 | 1800 | 641.4 | 1800 |
| 9 | 1800 | 1800 | 890.9 | 1800 | 300.1 | 1800 | 185.0 | 1800 |
| 10 | 1800 | 1800 | 841.5 | 1800 | 842.1 | 1800 | 215.3 | 1800 |
| Survival | 8/10 | 10/10 | 4/10 | 10/10 | 1/10 | 10/10 | 0/10 | 10/10 |

### 4.12. Model Test Results on Damage Case 4 with and without Deployment of a Watertight Barrier on the Trailer Deck

In Damage Case 4, the ship with GM 3.30 m capsized without mitigation rapidly in all cases with both damage opening times of 15 s and 30 s.

The mitigation effort with a rapidly deployable watertight barrier against floodwater spreading on the trailer deck of the MT RoPax was also tested in model scale with a physical model. The spreading of the floodwater on the trailer deck was prevented with the barrier at the building frame # 81, exactly as in the numerical simulations before (see Figure 16). Due to model construction, the floodable compartment volumes in the ship model show some deviations from those in the numerical model (see Chapter 4.8).

The ship with GM 3.30 m survived in calm water in all test cases with flooding mitigation, that is, in 100% of the cases, with both damage opening times (DOT) of 15 s and 30 s. Without mitigation, the ship capsized rapidly in all cases (see Figures 19 and 20).

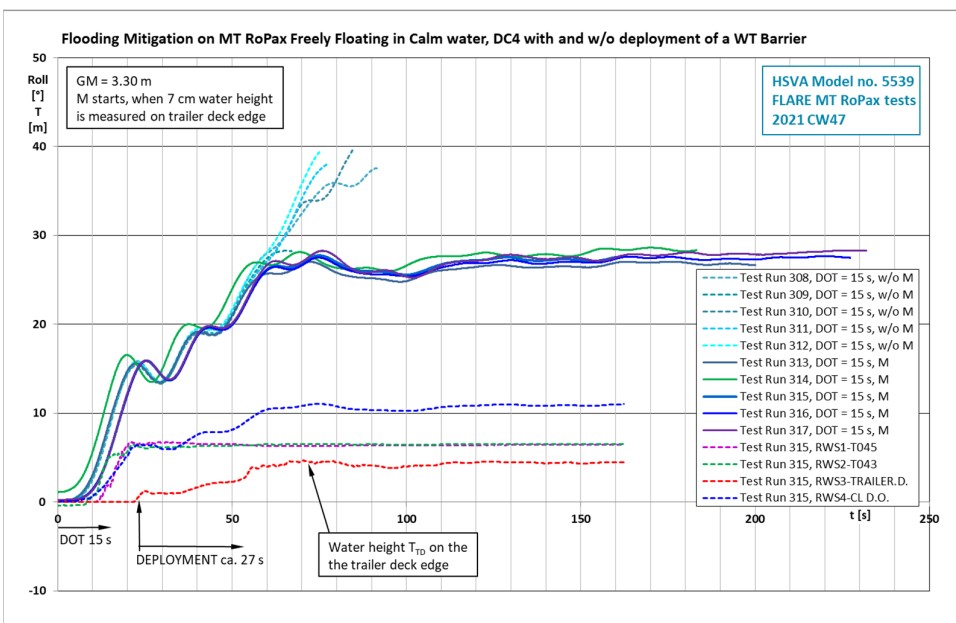

**Figure 19.** Damage Case 4 with damage opening time 15 s: Roll angle as a function of time with (solid lines) and without (dashed lines) deployment of a watertight barrier on the trailer deck. The dashed curves below show the water height in the damaged compartments, on the trailer deck, and just outside the damage opening for Test Run 315. Water height $T_{TRD}$ of 7 cm on the trailer deck starts the deployment of the WT barrier (RSW = relative wave sensor).

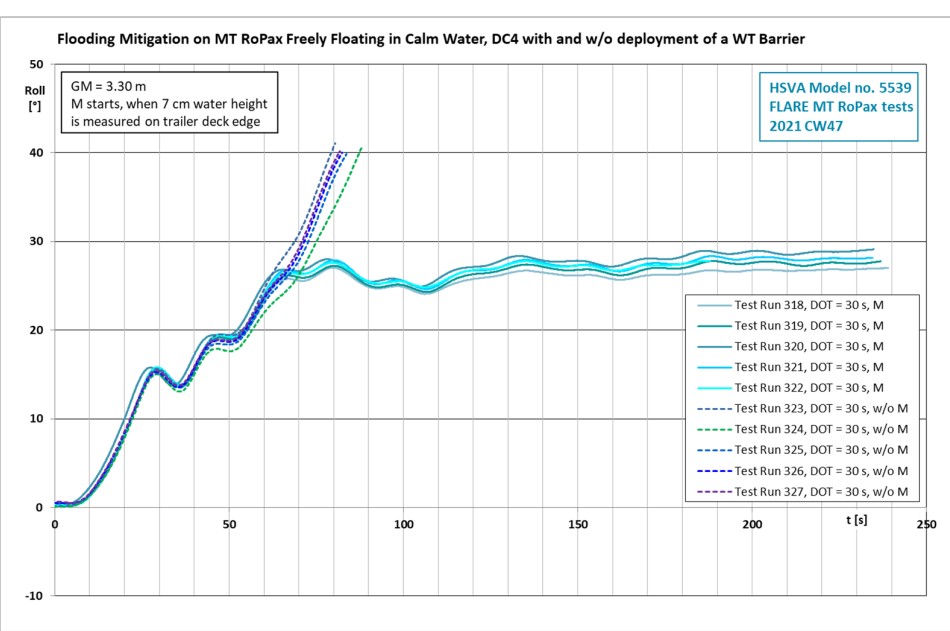

**Figure 20.** Damage Case 4 with damage opening time 30 s: Roll angle as a function of time with (solid lines) and without (dashed lines) deployment of a watertight barrier on the trailer deck.

With GM 3.71 m, the ship survived with flooding mitigation in irregular beam seas with $H_s$ 2.5 m in 100% of the cases when the damage was opened in 15 s. Without mitigation, the ship capsized in all cases. With $H_s$ 4.0 m, the ship survived with flooding mitigation in 100% of the cases when the damage was opened in 15 s. Without mitigation, the ship capsized in all cases, as before.

*4.13. Numerical Hindcast of the Flooding Mitigation with Deployment of a Watertight Barrier*

The a priori numerical simulations were used to define the parameters for the model tests. After these, a hindcast with more accurate and improved numerical simulation was carried out. The compartment volumes exactly as in the physical ship model were used in the hindcast. The use of the dynamic orifice equation instead of Bernoulli's equation for the inflow was not relevant in this case. All compartments below the trailer deck were modeled with the pendulum model, which is suitable for simple modeling of the complicated compartment geometries at hand. The oscillations in all experimental curves in Figure 21 show more water dynamics presence in the experiments than the numerical hindcast does. The correlation between the model test results and the numerical simulation in Figure 21 in the shown time domain is barely satisfactory, but the measured and computed curves converge to practically the same steady heeling angle value. The water ingress into the compartments below the trailer deck and onto the trailer deck is complicated, as the damage penetration was fully cut into the physical model (see Figure 16). As with the recovery of lost buoyancy in the same damage case, also in this mitigation case the experimental and numerical results were achieved with different GM values. Based on one hand on the measured steady heeling angle and the water height on the trailer deck edge shown in Figure 19, and on the other hand the computed water volume on the trailer deck, it can be concluded based on a simple volumetric calculation that considerably more water flows onto the trailer deck in the experiments than in the computations. Thus, the balance in the heeling angle in the experiments comes into being at a higher GM value than in the computations. As already mentioned in Chapter 4.9, for the same damage case, the numerical modeling shows weaknesses in modeling the rapid flooding of the complicated compartments in the damage area.

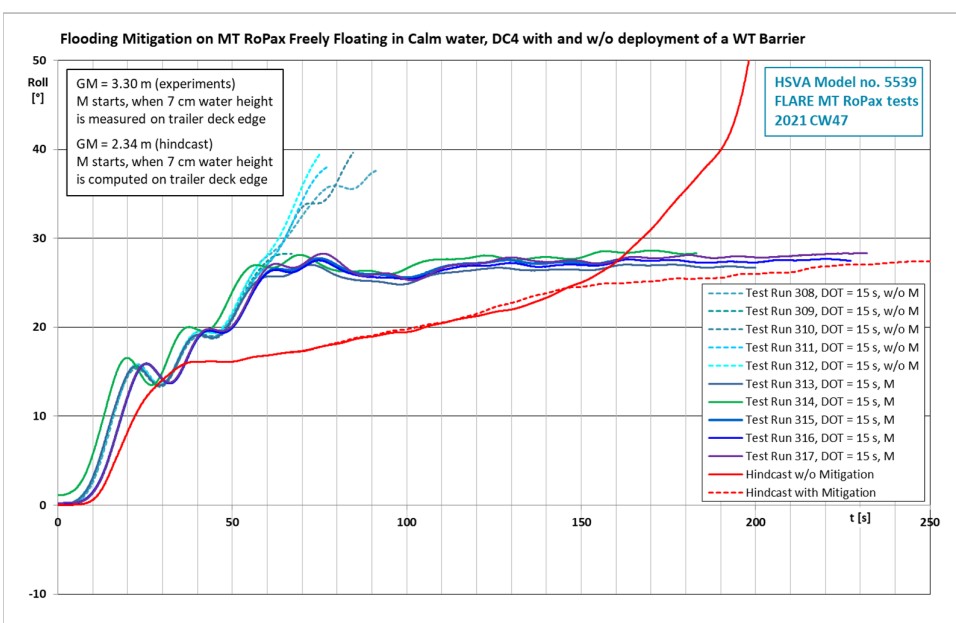

**Figure 21.** Damage Case 4 with damage opening time 15 s: Roll angle as a function of time with (solid lines) and without (dashed lines) recovery of lost buoyancy in the damaged compartment T045. The red solid line shows the numerical hindcast without mitigation, the dashed red line the hindcast with mitigation.

### 4.14. Conclusions on Damage Case 4 with and without Deployment of a Watertight Barrier

- The deployment of the watertight barrier on the trailer deck is a very straightforward flooding mitigation measure and can easily be installed on new or existing RoPax ships. The deployment time 30 s is based on built roll gates in use at HSVA. Before the barrier descended in the mitigation tests, some floodwater had already passed its position and had spread further onto the trailer deck. Regardless of this, the mitigation gives good results;
- If the capsize mechanism is gradual flooding, a short deployment time is certainly beneficial, but it does not appear to be crucial: in a few test computations with 30 s and 180 s deployment times, no significant differences in the results were found. Such a difference can be expected, when one or more high waves hit the breach on the ship hull just when it has opened. In a ship-to-ship collision, however, the damaged bow of the other ship would be just outside of the breach on the struck ship, between incoming waves and the breach. Thus, a sheltering effect against incoming waves can be expected;
- An earlier start of the deployment is of course beneficial. There is no practical reason that would prevent such roll gates to be lowered before the damage opens, letting a massive amount of floodwater in. The mitigation tests carried out demonstrate that also a fully automatic start of the mitigation based on floodwater detected on the trailer deck functions very well;
- The weakness of the method is that it does not recover anything, reduce heeling angle, or increase freeboard at damage location. It just stops the flooding progressing further on the important trailer deck. The heeling angle, as in this damage case, can remain high, and any disembarkation, if needed, would remain difficult in these conditions;
- Altogether, the curtain barrier remains an interesting, possibly cost-efficient solution, which can be added also to existing ships with relative ease. Instead of using a sensor to detect water on the vehicle deck, also other trigger/release mechanisms can be con-

sidered: (1) a general closure of WT doors, including the deployment of curtain barriers as a precautionary measure in case of a potential flooding hazard, (2) or at the latest, at the very onset of an accident, such as a collision.

## 5. An Overview of the Mitigation Methods

Three typical time histories of the roll angle representing flooding mitigation with counter flooding, with recovery of lost buoyancy, and with deployment of a watertight barrier on the trailer deck, are shown in Figure 22. As the three mitigation methods were investigated using two damage cases with different values of ship metacentric height GM, the comparison between these different mitigation methods is a more qualitative than quantitative one. Regardless of this, Figure 22 illustrates well the rapid reducing effect of the counter flooding on the heeling angle (blue curve), the much slower effect of the recovery of the lost buoyancy (violet-red), and the heeling angle remaining rather high in case of the deployment of the watertight barrier (deep green curve). These features are typical to the investigated mitigation methods.

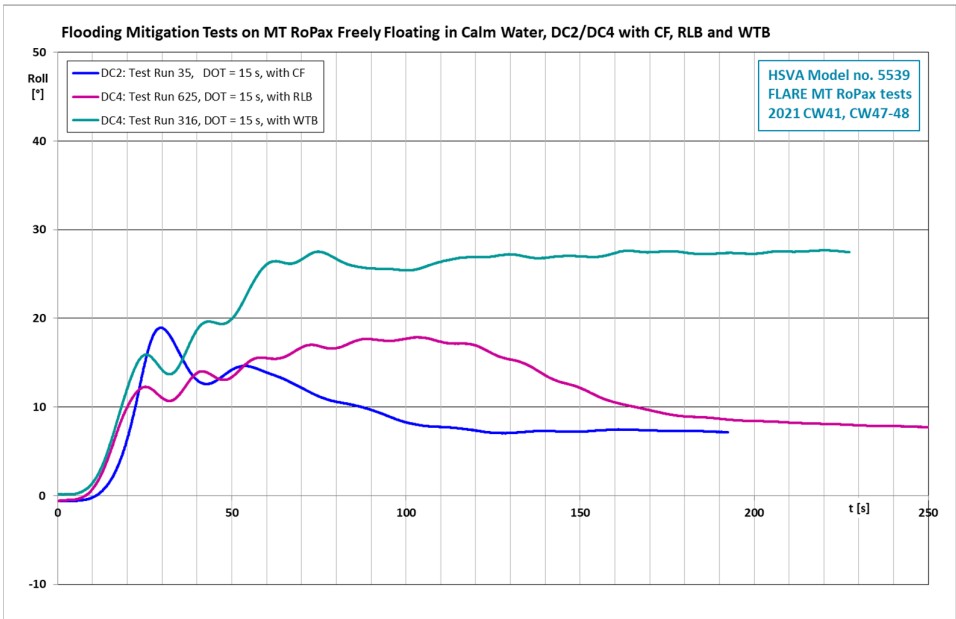

**Figure 22.** Qualitative comparison of the three different flooding mitigation methods on two damage cases: counter flooding (CF), recovery of the lost buoyancy (RLB), and deployment of the watertight barrier (WTB).

Tables 5–7 list the numerical and experimental mitigation results in the two studied damage cases in irregular beam seas in a concise manner. The arrows in the tables visualize the effect of the mitigation on the ship survivability in each case. As the tables show, all tested mitigation methods improve the ship survivability in these two damage cases on a certain range of wave heights. Due to unavoidable small discrepancies in modeling, the numerical simulations and the model test results show some differences, but they all very clearly demonstrate the benefits of the three flooding mitigation methods in the damage cases investigated.

For the studied damage cases, the different mitigation methods show some differences in their effectiveness, which depends also on the size and location of the damage, on those of the compartments available for counter flooding, or on the damaged compartment available for recovery of lost buoyancy through displacement of flood water with another lighter material, e.g., with expanding foam.

Both the counter flooding and the recovery of lost buoyancy in a damaged compartment improve the ship's condition, mainly by reducing the ship list. Deploying a watertight barrier on a trailer deck does not do this: it just prevents the ship stability from further deterioration by limiting the floodwater volume on the trailer deck. A large remaining heeling angle tends to be an unavoidable consequence. In view of many such aspects, any ranking of the mitigation methods investigated here is considered somewhat premature. The choice of the best flooding mitigation method should depend on the suitability of the particular method for the ship and the costs for the technical system to facilitate it.

**Table 5.** Numerical (blue) and experimental (red) flooding mitigation results with counter flooding in Damage Case 2, $Tp$ = 10.0 s.

| DC2 | Calm | | | | $Hs$ 2.5 m | | $Hs$ 3.0 m | | $Hs$ 3.5 m | | $Hs$ 3.5 m | | $Hs$ 4.0 m | | $Hs$ 5.0 m | |
|---|---|---|---|---|---|---|---|---|---|---|---|---|---|---|---|---|
| GM [m] | 3.018 | | | | 3.4 | | 3.4 | | 3.4 | | 3.48 | | 3.4 | | 3.48 | |
| DOT [s] | 15 | | 30 | | 15 | | 15 | | 15 | | 15 | | 15 | | 15 | |
| CF | no | yes | no | yes | no | yes | no | yes | no | yes | no | yes | no | yes | no | yes |
| Survival rate | 0/5 | 5/5 | 0/6 | 5/6 | 5/10 | 10/10 | 1/10 | 8/10 | 0/10 | 6/10 | 2/10 | 10/10 | 0/10 | 1/10 | 0/10 | 10/10 |

**Table 6.** Numerical (blue) and experimental (red) mitigation results with recovery of lost buoyancy in Damage Case 4, $Tp$ = 10.0 s.

| DC4 | Calm | | | | $Hs$ 2.5 m | | $Hs$ 4.0 m | | $Hs$ 5.0 m | | $Hs$ 5.5 m | | $Hs$ 6.0 m | | $Hs$ 7.0 m | |
|---|---|---|---|---|---|---|---|---|---|---|---|---|---|---|---|---|
| GM [m] | 3.71 | | | | 3.81 | | 3.81 | | 3.4 | | 3.4 | | 3.4 | | 3.4 | |
| DOT [s] | 15 | | 30 | | 15 | | 15 | | 15 | | 15 | | 15 | | 15 | |
| RLB | no | yes | no | yes | no | yes | no | yes | no | yes | no | yes | no | yes | no | yes |
| Survival rate | 0/5 | 5/5 | 0/5 | 5/5 | 0/10 | 10/10 | 0/10 | 8/10 | 8/10 | 10/10 | 4/10 | 6/10 | 1/10 | 4/10 | 0/10 | 0/10 |

**Table 7.** Numerical (blue) and experimental (red) mitigation results with deployment of a watertight barrier in Damage Case 4, $Tp$ = 10.0 s.

| DC4 | Calm | | | | $Hs$ 2.5 m | | $Hs$ 4.0 m | | $Hs$ 5.0 m | | $Hs$ 5.5 m | | $Hs$ 6.0 m | | $Hs$ 7.0 m | |
|---|---|---|---|---|---|---|---|---|---|---|---|---|---|---|---|---|
| GM [m] | 3.3 | | | | 3.71 | | 3.71 | | 3.4 | | 3.4 | | 3.4 | | 3.4 | |
| DOT [s] | 15 | | 30 | | 15 | | 15 | | 15 | | 15 | | 15 | | 15 | |
| WTB | no | yes | no | yes | no | yes | no | yes | no | yes | no | yes | no | yes | no | yes |
| Survival rate | 0/5 | 5/5 | 0/5 | 5/5 | 0/10 | 10/10 | 0/10 | 10/10 | 8/10 | 10/10 | 4/10 | 10/10 | 1/10 | 10/10 | 0/10 | 10/10 |

## 6. Conclusions

- Flooding mitigation measures on a modern RoPax design were investigated with numerical simulation and model tests in calm water and in irregular beam seas in two damage cases with and without the following mitigation efforts: (1) counter flooding; (2) recovery of lost buoyancy by displacing floodwater in a damaged compartment; and (3) deployment of a watertight barrier to prevent floodwater spreading on the large open trailer deck of the RoPax ship;
- The choice of the compartments for active flooding mitigation measures is based on the principles: (1) to provide righting moment to reduce ship list due to damage, (2) to maintain a slight trim (slope down) on the trailer deck towards the damage opening, (3) not to flood any compartments essential for the ship functions, and (4) to displace water only in compartments in which a foam system (or equiv.) can be applied, e.g., a potable water tank may not be suitable;

- The investigation throws light on the applicability and effectiveness of the investigated mitigation methods in calm water and in different beam sea conditions. Information on the ship stability, damage, and wave parameters suitable for model testing was generated;
- Damage Case 2 was investigated with and without counter flooding (CF). First, some computations with different duct opening sizes to the counter flooding compartments were carried out to show how the mitigation works in ideal calm water conditions with suitable ship GM values;
- Damage Case 4 was accordingly investigated with numerical simulations with and without the recovery of the lost buoyancy (RLB) by displacing floodwater in a damaged compartment, and also with and without the deployment of a watertight barrier (WTB) on the trailer deck to prevent further flooding;
- Although the numerical computations and model test results show some differences, they all very clearly demonstrate the benefits of the three flooding mitigation methods in the damage cases investigated;
- All the three mitigation methods investigated were found to be effective in either preventing or postponing ship capsize, thus providing good potential for the clear improvement of ship survivability in foreseeable damage cases. The mitigation methods studied are suitable for new and existing ships. This applies both to sudden flooding cases with transient floodwater and ship behavior as well as to gradual flooding cases;
- The GM values used in the tests deviate from the lowest GM value on the limit curve 3.4 m because GM values suitable for demonstrating the mitigation efforts were used in the computations and model tests. Using the GM value 3.4 m in all tests would have meant that a less clear demonstration of the effects would have been obtained, while in some cases the mitigation would not have been necessary, and in some cases, it would not have been sufficient;
- Planning the flooding mitigation for a ship design with a given GM would involve defining the extent of the mitigation methods that are sufficient to reach a desired survivability level, e.g., at given sea state. This is certainly possible in view of the information gained through the computations and model tests of the present study;
- Numerical simulations are required to design a flooding mitigation system. The simulation code should be able to handle different damage opening times and various release mechanisms for mitigation, e.g., the heeling angle or water depth in a compartment, and adjustable time delays in mitigation set-up. The programming effort for such amendments to an existing simulation code is small;
- The tested mitigation methods improve the ship survivability in these two damage cases on a certain range of wave heights. At low sea states, mitigation is often not needed. At the rarer high sea states, the effects of mitigation are not always sufficient to secure survival. In between, there is a range of sea states in which all the tested mitigation efforts considerably increase the ship survivability;
- In the simulations, the damage opening time of 15 s was mostly used. This is based on the assumption of the colliding ship pulling itself back at full power astern with its damaged bow withdrawing out of the collision damage penetration in the other ship. Presently, this is the best estimate of the shortest possible damage opening time. In a real case, the chances that the opening time would be longer should be rather high. This means that the calculated results should be conservative. That is, the mitigation systems are likely to be more effective and more largely applicable to a larger variety of cases than what the simulations and model tests here show;
- Counter flooding was found to be the easiest and fastest way to stabilize the damaged RoPax ship under study. Counter flooding has the advantage that few best suitable compartments on each side of the ship can be prepared for this. Their ship stabilizing effect can be applied to a large variety of damage cases on the opposite side of the ship.

- The recovery of lost buoyancy in the damaged compartment in principle implies the preparation of each compartment that can get damaged. At least preparation of the largest compartments far away from the centerline is necessary;
- The typical capsize mechanism of a RoPax ship involves the flooding of the trailer deck. Once this has progressed far enough, a capsize in waves is only a matter of limited time. The deployment of a watertight barrier on the trailer deck effectively prevents this, and the ship can survive in a damaged condition also in very high sea states. As there is no recovery in the damage extend, but just limitation, and the heeling angle can remain high, which has an adverse effect on all actions onboard. However, once the vessel is not in acute danger, the crew can concentrate on further stabilizing the ship.

**Funding:** The research presented in this paper was carried out in the framework of the project Flooding Accident Response (FLARE), no. 814753, under the H2020 program funded by the European Union, which is gratefully acknowledged. The views set out in this paper are solely those of the author.

**Institutional Review Board Statement:** Not applicable.

**Informed Consent Statement:** Not applicable.

**Data Availability Statement:** Not applicable.

**Acknowledgments:** The research presented would have hardly been possible without the framework of the EU project FLARE and it has certainly benefited from the open scientific communication between various distinguished colleagues in this framework and also outside of it. The following persons in HSVA have contributed to this study with the experiments carried out: K. Jacobsen, Y. Hong, P. Soukup, A. Schumacher, A. Beck, and T. Wirth.

**Conflicts of Interest:** The author declares no conflict of interest.

## Appendix A. Model Testing Techniques Applied in the Flooding Mitigation Tests

*Appendix A.1. Damage Opening Mechanisms*

The breaches on the hull related to Damage Cases 2 and 4 were opened with controllable speeds corresponding to the damage opening times of 15 s and 30 s in f.sc. Froude similarity is assumed. Thus, the Froude scale ratio for time is $\lambda^{1/2}$, in which $\lambda$ is the model scale factor. In Damage Case 2, two sliding doors moving horizontally in a purpose-built frame structure were used. Figure A1 illustrates the opening mechanism: The pulling strings are used to open the sliding doors at desired speed. The elastic cords pull the doors closed once the tension in pulling strings is released.

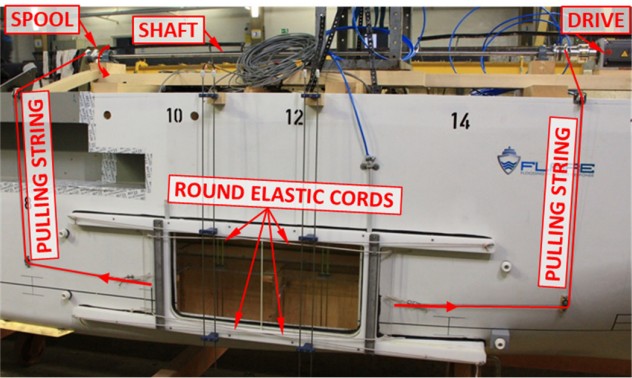

**Figure A1.** Damage opening mechanism for Damage Case 2.

The breach on the hull in Damage Case 4 lies at the ship shoulder, where the ship shell has curvature both in horizontal and vertical sections, as shown in Figure A2. The

horizontal curvature is adopted in the damage opening frame. The curvature in the vertical direction is large and the frame follows the ship shell closely. A flexible sliding door moving vertically and adjusting to the frame curvature in the vertical section was used in this case. In both damage cases, the sliding doors were moved with strings spooled on drums.

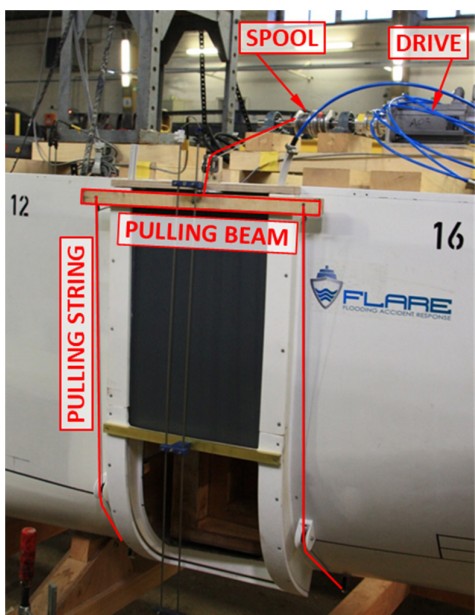

**Figure A2.** Damage opening mechanism for Damage Case 4.

*Appendix A.2. Dynamic Inflow into Counter flooding Compartments*

The counter flooding was investigated for Damage Case 2 in model tests after prior numerical simulations. Figure A3 illustrates the layout of the compartments and inflow ducts in the ship model. Due to constructional reasons in the scale model, the ducts to the counter flooding compartments had lengths of about 16–17 m in full scale. The counter flooding compartments were opened to sea using two pneumatic valves, located just at the limiting bulkheads of the counter flooding compartments. After the valves were once opened, they were kept open regardless of the further heeling motions of the ship.

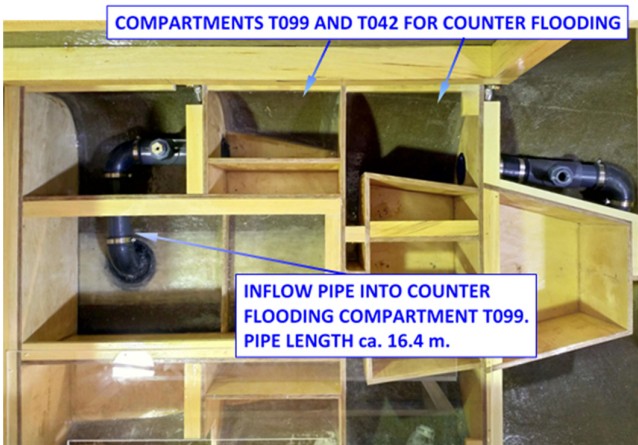

**Figure A3.** Counter flooding compartments T099 and T042 shown together with the inflow pipes. The damage opening on an acryl glass deck is partly visible at the lower edge of the photograph.

*Appendix A.3. Recovery of Lost Buoyancy in a Damaged Compartment by Displacing Flood Water*

The mitigation effort consisted of displacing floodwater in the compartment T045 on the damaged starboard side of the ship, or in other words, of the recovery of lost buoyancy in it. This was realized in the model tests with an inflatable container placed in the damaged compartment T045, as shown in Figure A4. The compartment has a complicated form and creating a suitable inflatable container for this purpose was not entirely trivial. For technical reasons in the model tests, it was possible to displace the floodwater volume of only 483 m$^3$ in full scale out of the total compartment volume of 793 m$^3$ of the compartment T045.

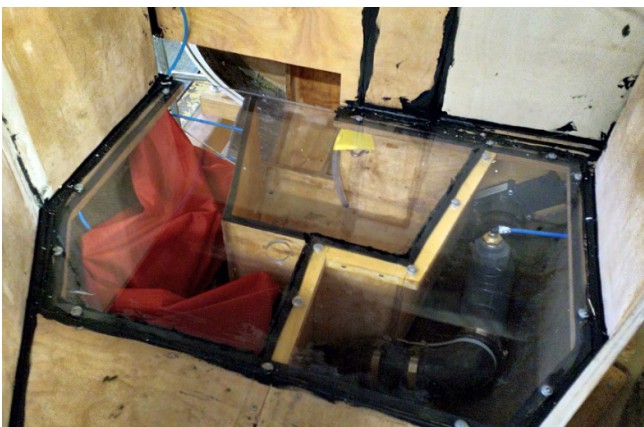

**Figure A4.** The red inflatable container for displacement of floodwater in compartment T045 during the assembly.

At the onset of mitigation, the filling of the container with compressed air was started, and the partly filled container rose and floated on top of the floodwater coming in. This somewhat delayed the mitigation effect. As the container inflated more, it started to displace floodwater out of the compartment as planned, reaching its maximum volume in a planned timeframe of 180 s (see Figure A5).

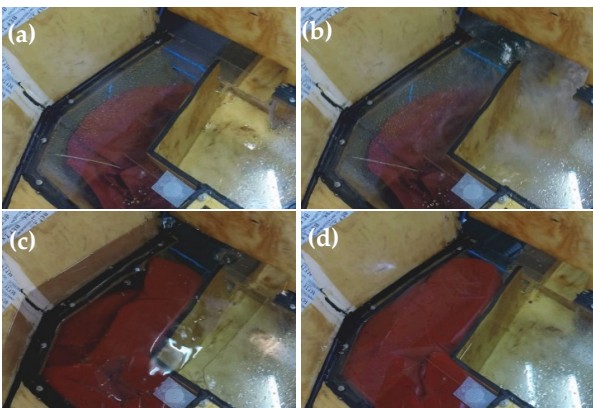

**Figure A5.** The red inflatable container for displacement of floodwater in compartment T045: (**a**) empty container in the empty compartment, (**b**) water rushes into the compartments, (**c**) the container being filled with pressurized air floats on the floodwater, (**d**) the fully inflated container has displaced most of the floodwater in the compartment.

*Appendix A.4. Deployment of a Watertight Barrier on the Trailer Deck*

A further test series of a mitigation effort with a rapidly deployable watertight barrier against floodwater spreading on the trailer deck of the MT RoPax was carried out in model scale with the physical ship model. The deployable curtain barrier on the trailer deck of the RoPax ship was realized in the physical scale model with an acryl glass sheet moving vertically in a housing placed on the trailer deck. The smooth descending motion of the acryl glass barrier was realized with a linear actuator with a spindle drive (see Figure A6).

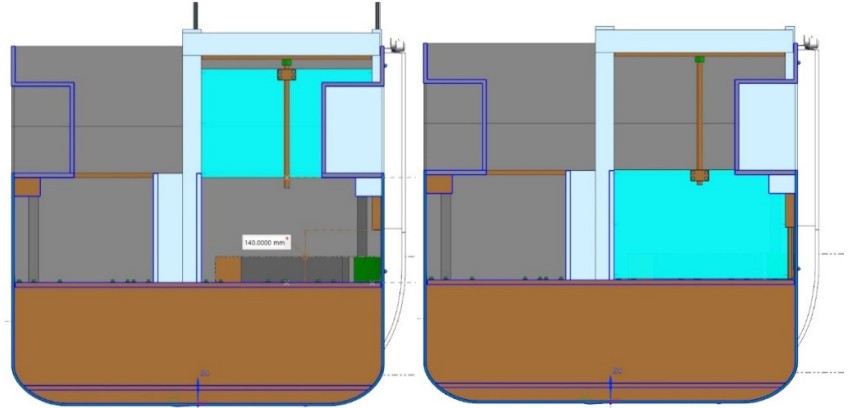

**Figure A6.** The deployable watertight barrier, shown with blue color: Left figure: high position, free water flow on the trailer deck. Right figure: the deployed barrier forms a watertight bulkhead on the trailer deck.

The barrier was deployed in the model tests with a duration of ca. 27 s describing the assumed deployment time needed to lower the curtain barrier in a ship. The barrier deployment started automatically as soon as the water depth of 0.07 m was reached at the water level sensor on a chosen location along the outer, starboard-side edge of the trailer deck between the barrier and the damage location (see Figure A7).

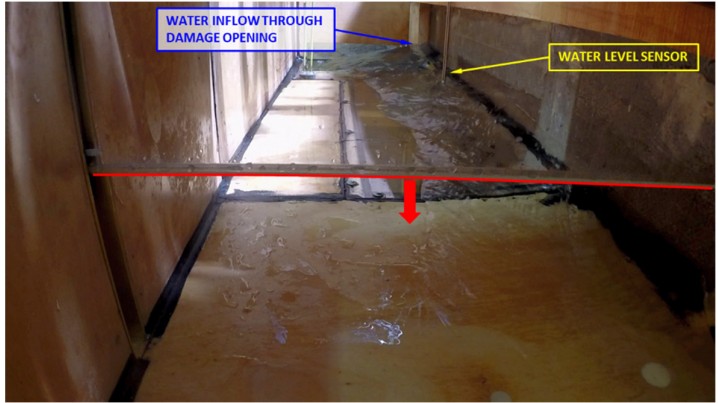

**Figure A7.** Camera view on the trailer deck. The positions of the damage opening and the water level sensor are indicated with arrows. The red line and arrow show the movement of the watertight barrier made of clear transparent acryl glass.

*Appendix A.5. Damage Opening Time and Start of the Flooding Mitigation*

In transient flooding cases, the damage opening time is an important factor influencing the outcome of the flooding and its mitigation. Realistic damage opening times were applied in the model tests. These were realized with simple adjustment of the rotating speed of the drums spooling the string lines pulling the sliding doors open.

Different release mechanisms for the flooding mitigation efforts were applied. The counter flooding was started automatically when a pre-set value of the ship heeling angle was exceeded. The recovery of lost buoyancy was started once a pre-set time delay after the damage opening had elapsed. The deployment of the watertight barrier on the trailer deck started automatically once water was detected by a sensor on the trailer deck. All release mechanisms were very simple and functioned without any problems throughout the test campaign.

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
