# Peer review of "Active Flooding Mitigation for Stability Enhancement in a Damaged RoPax Ship"

_jmse, doi:10.3390/jmse10060797_

Round 1

Reviewer 1 Report

The paper considers a fundamental issue related to Roro ship safety. The number and the continuous development of such ships makes the paper actual and most interesting.

The paper is very well written and presented. In some parts a few more tables could be useful to summarize the results for easier reading, (i.e. lines 143-153) but the figures help adequately.

A comment about the paper is the missing of some more systematic information about model test. Results of model tests are always presented in full scale. Some uncertainties and differences from model tests to full scale and more important to software simulation are highlighted, but in some cases this generates uncertainties about part of the reported results. I.e. at line 594 the phrase “the comparison is a more qualitative than quantitative one” could be explained in a comment to the experimental procedure as this last concerns geometrical and also dynamic aspects. I.e. at line 728 “They were opened with controllable speeds corresponding to the damage opening times of 15 s and 30 s” could be rephrased in terms of time giving the information of the correlation used. (If the given times refer to the ship, as presumably they are).

A further comment that could deserve an explanation by the Author is the choice of validating software results by scale model tests. In this case there are two sources of uncertainties, one due to the difference of methods and one other due to the full scale correlation. No uncertainty analysis is reported except some generic comments as (line 604): “The a-priori numerical simulations and the succeeding model test results show some differences.”

Author Response

I have kept my introduction short, a good introduction is given in [1], which is a 25 pages long open access publication. References directly related such an investigation as this were not found. This paper is stricktly about mitigation, not of model tests and not of numerical simulation. I have improved the wording in proper places related to the issues raised by you. I hope this makes it easier to undertand what I mean. All changes in the attached DPF are with red. Thank you for the good review, it has improved the paper.

Reviewer 2 Report

Read The Review Inform Paper jmse 1754583

Author Response

I have kept the introduction short and cited the ref. [1], which gives a good 25 pages long introduction to ship stability enhancement. References directly related to studies like the present one were not found. The topic of flooding mitigation is very new. I have improved the wording of some sentences, which may have been unclear. All changes in the attached PDF-file are marked with red. Thank you for your review.
